# A Temporally Correlated Latent Exploration for Reinforcement Learning

## Abstract

Efficient exploration remains one of the longstanding problems of deep reinforcement learning. Instead of depending solely on extrinsic rewards from the environments, existing methods use intrinsic rewards to enhance exploration. However, we demonstrate that these methods are vulnerable to *Noisy TV* and stochasticity. To tackle this problem, we propose Temporally Correlated Latent Exploration (TeCLE), which is a novel intrinsic reward formulation that employs an action-conditioned latent space and temporal correlation. The action-conditioned latent space estimates the probability distribution of states, thereby avoiding the assignment of excessive intrinsic rewards to unpredictable states and effectively addressing both problems. Whereas previous works inject temporal correlation for action selection, the proposed method injects it for intrinsic reward computation. We find that the injected temporal correlation determines the exploratory behaviors of agents. Various experiments show that the environment where the agent performs well depends on the amount of temporal correlation. To the best of our knowledge, the proposed TeCLE is the first approach to consider the action-conditioned latent space and temporal correlation for curiosity-driven exploration. We prove that the proposed TeCLE can be robust to the Noisy TV and stochasticity in benchmark environments, including Minigrid and Stochastic Atari.

## 1 Introduction

Reinforcement learning (RL) agents learn how to act to maximize the expected return of a policy. However, in real-world environments where rewards are sparse, agents do not have access to continuous rewards, which makes learning difficult. Inspired by human beings, numerous studies address this issue through intrinsic motivation, which uses so-called *bonus* or *intrinsic reward* to encourage agents to learn environments when extrinsic rewards are rarely provided (Schmidhuber, 1991b; Oudeyer & Kaplan, 2007a; Schmidhuber, 2010).

A notable intrinsic motivation is the curiosity-driven exploration method that adopts prediction error as intrinsic rewards (Oudeyer & Kaplan, 2007b; Pathak et al., 2017). For instance, Pathak et al. (2017) uses the difference between predicted states from the forward dynamics model and actual states as intrinsic rewards. Besides, the difference between the output of the fixed randomly initialized target network and the prediction network is adopted as intrinsic rewards (Burda et al., 2018b). Since the above methods encourage the exploration of rarely visited states, they can be useful in sparse reward environments such as Montezuma's Revenge (Mnih et al., 2015). However, curiosity agents can be trapped if the state prediction is inherently impossible or difficult. The problem of trapped agents could be caused by noise sources such as the Noisy TV or stochasticity in environments (Burda et al., 2018b; Pathak et al., 2019; Mavor-Parker et al., 2022). Therefore, it is challenging for curiosity agents to learn environments where noise sources exist.

To overcome the limitation, this paper proposes **Temporally Correlated Latent Exploration** (TeCLE), a novel curiosity-driven exploration method that employs an action-conditioned latent space and temporal correlation. Firstly, this paper formulates intrinsic reward from the difference between the reconstructed states and actual states. Secondly, we introduce the conditioned latent spaces for exploration. Whereas previous studies (Oh et al., 2015; Kim et al., 2018) use action as a condition for prediction problems, the proposed TeCLE uses action as a condition variable to learn a conditioned latent space, which is referred to as *action-conditioned latent space*. In the

proposed method, the state is embedded as a state representation, which is then encoded into an action-conditioned latent space. This enables the action-conditioned latent space to learn the distribution of the state representation, allowing agents to effectively avoid noise sources. Whereas previous works have used conditioned latent spaces to alleviate the out-of-distribution (OOD) problem in offline RL (Zhou et al., 2021; Rezaeifar et al., 2022), this paper employs the conditioned latent space for curiosity-driven exploration methods. On the other hand, temporal correlation using colored noise was successfully applied to the action selection for RL agents (Eberhard et al., 2023; Hollenstein et al., 2024). Different from the above works, our proposed method injects temporal correlation into the action-conditioned latent space. As far as we know, this paper is the first approach to inject temporal correlation for intrinsic motivation. To prove the effectiveness, we evaluate our proposed TeCLE on Minigrid and Stochastic Atari, comparing its performance with baselines. Furthermore, the generalization ability of TeCLE is demonstrated through experimental results with no extrinsic reward setting. For a more qualitative analysis, we discuss the performance that depends on the amount of temporal correlation (i.e., colored noise) and propose an optimal colored noise according to the properties of the noise source and the environment. The contributions of our study are summarized as follows:

- **Defining Intrinsic Rewards via Action-Conditioned Latent Spaces:** Since the action-conditioned latent space reconstructs states by learning the distribution of states, it avoids being trapped in noise sources where the state prediction is inherently impossible. Therefore, we formulate intrinsic rewards using action-conditioned latent spaces for exploration.

- **Introducing Temporal Correlation for Intrinsic Motivation:** By injecting colored noise into the action-conditioned latent space, we further introduce temporal correlation into the computation of intrinsic reward. Furthermore, we find that different colors of noise encourage agents to have different exploratory behaviors.

- **Benchmarking the Performance:** To evaluate the effectiveness of the proposed TeCLE, we conduct extensive experiments on the Minigrid and Stochastic Atari environments. Compared to several strong baselines, TeCLE achieves good performance not only on difficult exploration tasks but also on environments where noise sources exist.

## 2 RELATED WORKS

### 2.1 EXPLORATION WITH INTRINSIC MOTIVATION

The *bonus* or *intrinsic reward* in RL refers to an additional reward often used to encourage exploration of less frequently visited states. In the count-based exploration method, state-action visitation was directly used to compute intrinsic reward (Strehl & Littman, 2008). To reduce computational efforts and generalize intrinsic rewards to a large state-space, numerous works have been studied (Bellemare et al., 2016; Martin et al., 2017; Ostrovski et al., 2017; Tang et al., 2017; Choshen et al., 2018; Choi et al., 2018; Machado et al., 2020). However, the above count-based methods can be less effective in sparse reward environments and break down when the number of novel states is larger than their approximation (Raileanu & Rocktäschel, 2020; Mavor-Parker et al., 2022).

On the other hand, curiosity-based exploration method proposed to predict the dynamics of the environment to compute intrinsic reward (Schmidhuber, 1991a;b; Oudeyer & Kaplan, 2007a; Stadie et al., 2015). Using a self-supervised manner, the curiosity can be quantified as the prediction error or uncertainty of a consequence of the actions (Pathak et al., 2017; Burda et al., 2018a; Pathak et al., 2019; Raileanu & Rocktäschel, 2020). Moreover, Burda et al. (2018b) introduced a novel framework where the prediction problem is randomly generated. Whereas the above curiosity-driven exploration methods were effective on several sparse reward environments in Atari (Mnih et al., 2015), Noisy TV or stochasticity can misdirect the curiosity of the curiosity agent (Raileanu & Rocktäschel, 2020; Mavor-Parker et al., 2022).

### 2.2 TEMPORALLY CORRELATED NOISE AS ACTION NOISE

A common exploration technique in RL is to add noise such as Ornstein–Uhlenbeck (OU) noise (Uhlenbeck & Ornstein, 1930) or Gaussian noise to an action sampled from the policy. Recently, several studies introduced different types of action noise. Eberhard et al. (2023) studied the effects of the

temporally correlated noise as action noise for off-policy algorithms in continuous control environments. Besides, the amount of the temporal correlation, which depends on the color parameter $\beta$, was described as colored noise. The evaluation of different kinds and colors of noise shows that pink noise ($\beta = 1.0$), which has the intermediate amount of Gaussian noise ($\beta = 0$) and OU noise ($\beta \approx 2$), can be the optimal noise in action selection. Furthermore, Hollenstein et al. (2024) studied the effects of the temporally correlated noise for on-policy algorithms, where an intermediate amount of temporal correlation between Gaussian noise and pink noise with $\beta = 0.5$ achieved the best performance. However, there is no attempt to introduce temporal correlations to intrinsic motivation, in contrast to the action selection.

### 2.3 Conditional Variational Auto-Encoder (CVAE) for Exploration

CVAE (Sohn et al., 2015) was introduced to learn the unlabeled dataset efficiently. Since input variables are encoded as probability distributions into the conditioned-latent spaces, the policy of RL agents can be efficiently modeled. Thus, several studies adopted CVAE to mitigate the OOD problem in offline-RL. Zhou et al. (2021) employed CVAE to model the behavior policies of agents for a dataset or pre-collected experiences. The policy network was trained from the latent behavior space, and its decoder was used to output actions from the behavior space of the environment. Since the latent space after training was fit for the dataset distribution, the OOD problem of generating unpredictable actions could be mitigated. Besides, Rezaeifar et al. (2022) computed intrinsic reward for anti-exploration using the $L_2$-norm between the predicted action by a decoder and actual action. Unlike previous studies (Klissarov et al., 2019; Kubovčík et al., 2023; Yan et al., 2024) that adopted VAE for intrinsic motivation, numerous studies adopted CVAE to model the policy networks.

## 3 Background

In this paper, we use the Markov Decision Process (MDP) of a single RL agent represented as a tuple $\mathcal{M} = (\mathcal{S}, \mathcal{A}, \mathcal{P}, r, \gamma)$. The tuple includes a set of states $\mathcal{S}$, a set of actions $\mathcal{A}$, and the transition function $\mathcal{P} : \mathcal{S} \times \mathcal{A} \times \mathcal{S} \to [0, 1]$ that provides the distribution $\mathcal{P}(s'|s, a)$ over the next possible successor state $s'$ given a current state $s$ and action $a$. The agent chooses an action from a deterministic policy $\pi : S \to A$ and receives a reward $r : \mathcal{S} \times \mathcal{A} \to \mathbb{R}$ at each time step. The goal of the agent is to learn the policy that maximizes the discounted expected return $\mathcal{R}_t = \mathbb{E}[\sum_{k=0}^{t} \gamma^k r_{t+k+1}]$ at a time step $t$, where $\gamma \in [0, 1]$ is the discount factor and $r_t$ is the sum of the extrinsic reward $r_t^e$ and the intrinsic reward $r_t^i$, respectively.

Pathak et al. (2017) proposed Intrinsic Curiosity Module (ICM) to formulate future prediction errors as the intrinsic reward. Since making predictions from the raw states is undesirable, ICM uses an embedding network $f_\theta$ that takes the state representation $\phi(s_t) = f_\theta(s_t)$ by training the learnable parameters $\theta$ using two submodules as: firstly, the inverse dynamics model $g_\theta$ in the first submodule takes $\phi(s_t)$ and $\phi(s_{t+1})$ as its inputs. The inverse dynamics model $g_\theta$ predicts the action of agents $\hat{a}_t$, which is equated as $\hat{a}_t = g(\phi(s_t), \phi(s_{t+1}))$. Model $g_\theta$ is trained to minimize $L_I = CrossEntropy(\hat{a}_t, a_t)$ denoting the loss from the error between $\hat{a}_t$ and $a_t$. The forward dynamics model $h$ in the second submodule takes $\phi(s_t)$ and $a_t$ as its inputs. The forward dynamics model $h$ predicts the next state representation $\hat{\phi}(s_{t+1})$, which is equated as $\hat{\phi}(s_{t+1}) = h(\phi(s_t), a_t)$. Model $g_\theta$ is trained to minimize $L_F = ||\hat{\phi}(s_{t+1}) - \phi(s_{t+1})||_2^2$ denoting the loss from the error between $\hat{\phi}(s_{t+1})$ and $\phi(s_{t+1})$.

## 4 TeCLE: Temporally Correlated Latent Exploration

Although the existing curiosity-driven methods improved exploration, they can be vulnerable to Noisy TV problems or stochasticity of environments (Raileanu & Rocktäschel, 2020; Mavor-Parker et al., 2022). TeCLE started with the assumption that this is caused by predicting the noise sources, which is inherently impossible, and the predictions themselves must contain noise to solve this problem. In the following paragraphs, we describe the role and effect of each part. Consequently, in part C. Colored noise, we prove that these effects ultimately help the agents deal with the Noisy TV problem. As shown in Figure 1, TeCLE consists of three parts, and the intrinsic reward is computed

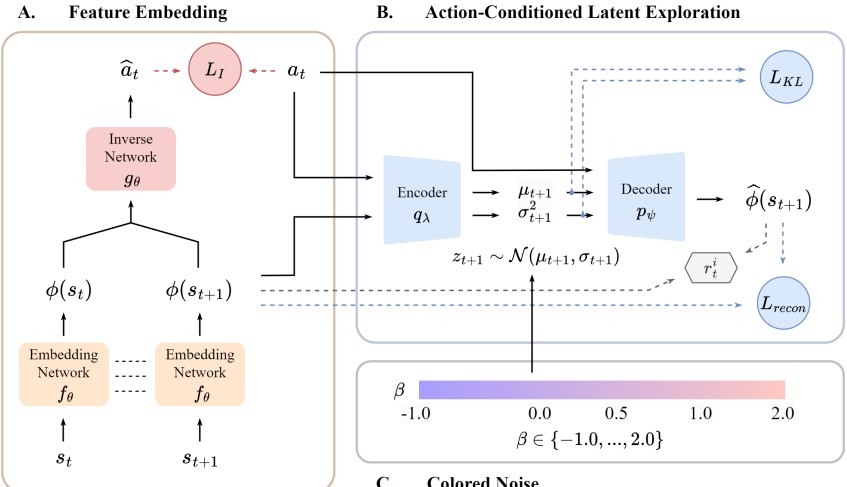

**Figure 1:** Architecture of proposed TeCLE. (Part A) **Feature Embedding** learns the state representations $\phi(s_t)$ and $\phi(s_{t+1})$ using embedding network $f_\theta$ and inverse network $g_\theta$; (Part B) **Action-Conditioned Latent Exploration** computes intrinsic reward $r_t^i$ using the reconstructed state representation $\hat{\phi}(s_{t+1})$ and the $\phi(s_{t+1})$; (Part C) **Colored Noise** injects $\varepsilon_{t+1}$ when sampling the latent representation $z_{t+1}$ of Part B.

separately from the policy networks. Similar to other curiosity-driven exploration methods, the intrinsic reward is computed separately from the policy networks.

## A. FEATURE EMBEDDING

It has been proven that predicting feature space leads to better generalization compared with predicting raw pixel space (Burda et al., 2018a). Furthermore, since predicting the raw pixel is challenging (Pathak et al., 2017), we use the embedding network and inverse network to learn the state representation. In our formulation, embedding network $f_\theta$ that shares the parameters takes states $s_t$ and $s_{t+1}$ as inputs. To optimize $f_\theta$, state representation $\phi(s_t)$ and future state representation $\phi(s_{t+1})$ are used as input of the inverse network $g_\theta$ as:

$$\hat{a}_t = g_\theta(\phi(s_t), \phi(s_{t+1})), \tag{1}$$

where $\hat{a}_t$ denotes the predicted action. The loss function $L_I$ is equated as:

$$L_I = CrossEntropy(\hat{a}_t, a_t). \tag{2}$$

By learning state representations through embedding networks, the agent extracts important information from the environment, such as things that agents can control (e.g., steering wheel) and things that agents cannot control but can be affected (e.g., passing vehicles). Detailed explanations of the state representation and inverse network are provided in Section 3.

## B. ACTION-CONDITIONED LATENT EXPLORATION

Several existing studies use the $\phi(s_{t+1})$ and the predicted future state representation $\hat{\phi}(s_{t+1})$ in the computation of the intrinsic reward (Pathak et al., 2017; Burda et al., 2018a; Pathak et al., 2019). Unlike the above approaches, intrinsic reward of the proposed TeCLE is computed by using the reconstructed $\phi(s_{t+1})$ from the action-conditioned latent space. Firstly, $\phi(s_{t+1})$ and action $a_t$ are taken as inputs of an encoder $q_\lambda$ as denoted in Eq.(3). Each corresponds to an input variable **x** and a condition variable $y$ of CVAE.

$$q_\lambda(z_{t+1}|\mathbf{x}, y) := q_\lambda(z_{t+1}|\phi(s_{t+1}), a_t), \quad z_{t+1} \sim \mathcal{N}(\mu_{t+1}, \sigma_{t+1}), \tag{3}$$

where latent representation $z_{t+1}$ is sampled using the $\mu_{t+1}$ and $\sigma_{t+1}^2$ from output of the encoder $q_\lambda$. Then, $z_{t+1}$ and $a_t$ are taken as inputs to the decoder $p_\psi$, which outputs $\hat{\phi}(s_{t+1})$ as:

$$p_\psi(\hat{\phi}(s_{t+1})|z_{t+1}, a_t). \tag{4}$$

Consequently, the intrinsic reward $r_t^{\mathrm{i}}$ is computed using $L_2$-norm of the difference between $\hat{\phi}(s_{t+1})$ and $\phi(s_{t+1})$ as follows:

$$r_t^{\mathrm{i}} = \|\hat{\phi}(s_{t+1}) - \phi(s_{t+1})\|_2. \tag{5}$$

The intuition for how TeCLE can encourage better exploration while avoiding noise sources is as follows: $p_\psi$ reconstructs the $\hat{\phi}(s_{t+1})$ based on the probabilities of the previously visited states. Besides, $a_t$ is used as a condition variable of the $q_\lambda$ and $p_\psi$ for self-supervised learning. Therefore, the proposed TeCLE can encourages agents to explore by assigning larger intrinsic rewards to rarely visited states in a self-supervised manner, while avoiding noise sources based on state visitation probabilities. The training loss is the sum of reconstruction loss $L_{recon}$ and KL divergence $L_{KL}$, where each loss function is formulated as:

$$L_{recon} = BinaryCrossEntropy(\hat{\phi}(s_{t+1}), \phi(s_{t+1})). \tag{6}$$

$$L_{KL} = KL(q_\lambda(z_{t+1}|\phi(s_{t+1}), a_t)||p_\psi(z_{t+1}|\phi(s_{t+1}))). \tag{7}$$

The detailed formulation and explanation of optimization are described in Appendix A.1.

## C. COLORED NOISE

It has been demonstrated that temporally correlated noise for action selection enhances exploration in both on-policy and off-policy RL (Eberhard et al., 2023; Hollenstein et al., 2024). However, as far as we know, there have been no attempts to apply temporal correlation to intrinsic motivation. Therefore, we consider the utilization of temporally correlated noise when computing the intrinsic reward. To explain the temporally correlated noise, we revisit $z_{t+1} \sim \mathcal{N}(\mu_{t+1}, \sigma_{t+1})$ in Eq.(3). Using a reparameterization trick, it can be re-written as:

$$z_{t+1} = \mu_{t+1} + \varepsilon_{t+1}\sigma_{t+1}, \tag{8}$$

where $\varepsilon_{t+1}$ is the injected noise. If $\varepsilon_{(1:t)} = (\varepsilon_1, \cdots, \varepsilon_i, \cdots, \varepsilon_t)$ is sampled from the Gaussian distribution at every timestep, any $\varepsilon_i, \varepsilon_j \in \varepsilon_{(1:t)}$ can be expressed as *temporally uncorrelated*. Besides, temporally uncorrelated noise (i.e., white noise) corresponds to color parameter $\beta = 0$. In terms of signal processing, $|\hat{\varepsilon}_{(1:t)}(f)^2|$ and $\hat{\varepsilon}_{(1:t)}(f)$ is converted as the Power Spectral Density (PSD) of $\varepsilon_{(1:t)}$ and the Fourier transform of $\varepsilon_{(1:t)}$, where $\beta$ has the properties of $|\hat{\varepsilon}_{(1:t)}(f)^2| \propto f^{-\beta}$ (Timmer & Koenig, 1995; Eberhard et al., 2023). Therefore, it can be concluded that $\beta$ controls the amount of temporal correlation in the $\varepsilon_{(1:t)}$. In other words, the noise with $\beta > 0$ produces a temporal correlation between any $\varepsilon_i, \varepsilon_j \in \varepsilon_{(1:t)}$ at different time steps. On the other hand, the noise with $\beta < 0$ produces a *temporal anti-correlation* between any $\varepsilon_i, \varepsilon_j \in \varepsilon_{(1:t)}$, causing high variation of noises between time steps. A more detailed explanation of colored noise sequences is described in Appendix A.2.

In our intrinsic formulation, the generated $\varepsilon_{t+1}$ is used to sample the latent representation $z_{t+1}$, and the $\hat{\phi}(s_{t+1})$ is reconstructed from $p_\psi$ using $z_{t+1}$ and $a_t$. Therefore, it can be considered that sequence $\hat{\phi}(s_{(1:t)}) = (\hat{\phi}(s_1), \cdots, \hat{\phi}(s_t))$ has an amount of temporal correlation, depending on $\beta$. We hypothesize that the temporal correlation and anti-correlation ($\beta \neq 0$) in the generated noise sequence determine the exploratory behavior of the agent. When temporally anti-correlated noise with $\beta < 0$ is injected, noise sequences with constantly fluctuating magnitude can dynamically produce the reconstructed state sequence. Thus, agents can be less sensitive to novel states, making them more robust to Noisy TV by assigning smaller intrinsic rewards than when $\beta \geq 0$. Besides, in the injection of temporally correlated noise with $\beta > 0$, the noise sequence with smooth changing magnitude generates a larger intrinsic reward in the novel states than when $\beta \leq 0$. To be more specific, temporally anti-correlated noise with $\beta < 0$ can make the proposed TeCLE continue to have a perturbation of subsequent intrinsic rewards. On the other hand, the smooth change of temporally correlated noise with $\beta > 0$ makes the change of subsequent intrinsic rewards stable. Therefore, we expect that TeCLE can achieve higher performance with temporally correlated noise ($\beta > 0$) in sparse reward environments and with temporally anti-correlated noise ($\beta < 0$) in environments where Noisy TV exists. However, since the reconstructed states are unstable at the beginning of the training due to the nature of the generative model (Regenwetter et al., 2022), the effect of the colored noise can be small. In other words, when the model is sufficiently trained, the effects of colored noise can be significant depending on $\beta$.

In the following section, we discuss this tendency of colored noise and prove our hypothesis. Furthermore, extensive experiments were conducted to observe the exploratory behavior of TeCLE with various colored noises. We also analyze the results to derive the optimal $\beta$ for each task.

## 5 EXPERIMENTAL RESULTS AND ANALYSIS

In the experiments, we analyzed the performance of TeCLE by varying $\beta$ of generated noise sequence $\varepsilon_{(1:t)}$. Also, we proved the effectiveness of TeCLE by comparing it with baselines in the Minigrid and Stochastic Atari environments. Further experiments, including the hard exploration tasks, can be found in the Appendix.

### 5.1 EXPERIMENTAL SETUP

**Baseline:** For all our experiments, we adopted Proximal Policy Optimization (PPO) (Schulman et al., 2017) as the base RL algorithm and Adam (Kingma, 2014) as the optimizer. In the experimental results, term *ICM* refers to the Intrinsic Curiosity Module, which uses the forward dynamics-based prediction error as the intrinsic reward (Pathak et al., 2017). Term *RND* refers to the Random Network Distillation, which uses the fixed randomly initialized network-based prediction error as the intrinsic reward (Burda et al., 2018b). Besides, terms *TeCLE (-1.0)* and *TeCLE (2.0)* refer to our proposed TeCLE with blue ($\beta = -1.0$) and red ($\beta = 2.0$) noises, respectively. All models used the same base RL algorithm and neural network architecture for both the policy and value functions. The only difference among them was in how intrinsic rewards were defined. Details on the hyperparameters and neural network architectures can be found in Appendix C. For the comparison, we adopted average return during training as the performance metric. In the experimental results, solid lines and shade regions of training results denote the mean and variance, respectively.

**Environments:** Since we focused on the exploration ability of agents, we not only used rewards but also directly measured the state coverage (Raileanu & Rocktäschel, 2020; Kim et al., 2023) for evaluation. In the Minigrid experiments, the world is partially observable (Chevalier-Boisvert et al., 2018). Also, $N \times N$ in the environment name refers to the size of a map, and $SXRY$ refers to a map of size $X$ with rows of $Y$. Besides, $SXNY$ refers to $X$ size map with $Y$ number of valid crossings across lava or walls from the starting position to the goal. Additionally, *Noisy TV* experiments were implemented by adding action-dependent noise when the agent selects *done* action in environments (Raileanu & Rocktäschel, 2020). In the Stochastic Atari experiments, we adopted sticky actions (i.e., randomly repeating the previous action (Burda et al., 2018b)), which were proposed by Machado et al. (2018).

### 5.2 DISCUSSION AND ANALYSIS OF EFFECTS OF DIFFERENT COLORED NOISE

**Table 1:** Normalized average returns according to $\beta$ in the Minigrid and Stochastic Atari environments with and without Noisy TV. Each value represents the average result from 3 seeds, with the best score in bold.

| | With Noisy TV | | | | | | Without Noisy TV | | | | |
|---|---|---|---|---|---|---|---|---|---|---|---|
| Environment | -1.0 | 0.0 | 0.5 | 1.0 | 2.0 | Environment | -1.0 | 0.0 | 0.5 | 1.0 | 2.0 |
| DoorKey8 × 8 | **.697** | .379 | .318 | .536 | .565 | DoorKey8 × 8 | .647 | **.839** | .713 | .689 | .771 |
| DoorKey16 × 16 | **.311** | .048 | .040 | .033 | .200 | DoorKey16 × 16 | .209 | .041 | **.294** | .019 | .286 |
| LCS9N3[1] | .921 | .930 | **.934** | .929 | .932 | LCS9N3[1] | .941 | **.941** | .941 | .939 | .940 |
| LCS11N5[1] | .000 | .000 | .000 | .000 | .000 | LCS11N5[1] | .485 | .000 | .000 | **.719** | .430 |
| DO8 × 8[2] | .536 | .929 | .884 | .903 | **.947** | DO8 × 8[2] | .730 | .300 | .691 | **.970** | .877 |
| DO16 × 16[2] | .631 | .959 | .954 | **.978** | .968 | DO16 × 16[2] | .819 | .807 | **.958** | .956 | .897 |
| Empty8 × 8 | **.939** | .936 | .938 | .938 | .938 | Empty8 × 8 | .935 | **.939** | .935 | .933 | .937 |
| Empty16 × 16 | .921 | .913 | .901 | .912 | **.927** | Empty16 × 16 | **.936** | .874 | .905 | .903 | .924 |
| KeyCorridorS3R3 | .000 | .000 | **.001** | .000 | .000 | KeyCorridorS3R3 | .079 | .524 | .000 | .156 | **.087** |
| MultiRoomN2S4 | .814 | .813 | **.815** | .813 | .814 | MultiRoomN2S4 | .827 | .827 | **.828** | .828 | .823 |
| BankHeist[3] | **.719** | .687 | .651 | .676 | .580 | SpaceInvaders | .420 | **.650** | .599 | .519 | .619 |

[1] LavaCrossing environment in Minigrid
[2] DynamicObstacles environment in Minigrid
[3] Natural Noisy TV environment in Atari (Mavor-Parker et al., 2022; Jarrett et al., 2023)

In this subsection, we performed experiments in environments with and without Noisy TV to show the exploratory behaviors of the TeCLE with various colored noises. To analyze the effects when temporally correlated noise is injected into action-conditioned latent

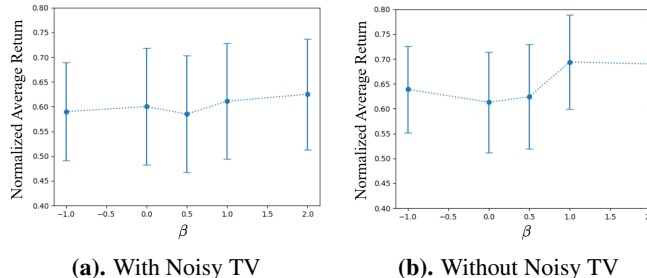

**(a).** With Noisy TV      **(b).** Without Noisy TV

**Figure 2:** Normalized average returns across environments in Table 1. The error bars show the mean (dots) and standard error (upper and lower bounds) of the normalized average returns according to $\beta$.

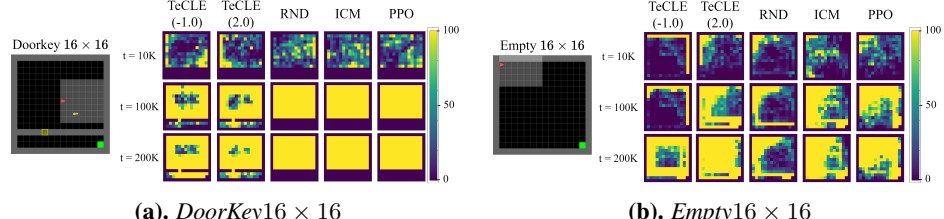

**(a).** *DoorKey*$16 \times 16$          **(b).** *Empty*$16 \times 16$

**Figure 3:** Visualized state coverages in *DoorKey*$16 \times 16$ and *Empty*$16 \times 16$ without Noisy TV. In *DoorKey*$16 \times 16$, only TeCLE with red ($\beta = 2.0$) and blue ($= -1.0$) noises can solve the tasks and learned the optimal policy for exploration. It seems that blue noise ($\beta = -1.0$) encourages agents to exploit more than explore compared to red noise. We think that TeCLE encourages the agent to explore more than exploit compared to low $\beta$, which is similar to the studies in Eberhard et al. (2023); Hollenstein et al. (2024).

space and find the optimal $\beta$ for each environment, experiments were performed with $\beta \in \{-1.0 \text{ (blue noise)}, 0 \text{ (white noise)}, 0.5, 1.0 \text{ (pink noise)}, 2.0 \text{ (red noise)}\}$ in $\varepsilon_{(1:t)}$ on the Minigrid and Stochastic Atari environments. Besides, the normalized average return (Hollenstein et al., 2024) was chosen as the performance metric.

In Table 1, when the blue noise ($\beta = -1.0$) was applied to the environments with Noisy TV, the normalized average returns in four environments had the highest values. Notably, compared with the cases applying the white noise ($\beta = 0$), the experiments for *DoorKey* environments significantly increased the normalized average returns. Additionally, the experiments with the red noise ($\beta = 2.0$) showed good normalized average returns. Overall, when averaging the normalized average returns across environments with Noisy TV, the experiments with the red noise ($\beta = 2.0$) produced the highest value, as shown in Figure 2 (a). On the other hand, white noise produced the highest value in the four environments without Noisy TV. However, experiments on *DoorKey*$16 \times 16$ and *DO*$8 \times 8$ with white noise ($\beta = 0$) showed significantly degraded results than other colored noises. Notably, the experiments with the red noise ($\beta = 2.0$) also showed good normalized average returns.

As we hypothesized in Section 4, experimental results demonstrate that the amount of temporal correlation is closely related to the robustness of the agent against the Noisy TV. The results in Table 1 show that blue noise ($\beta = -1.0$) achieves good normalized average returns compared to other noises in Noisy TV environments. This shows that blue noise ($\beta = -1.0$) learned the optimal policy faster than other colored noises while avoiding being trapped by the Noisy TV. On the other hand, red noise ($\beta = 2.0$) was generally more effective in all environments than other colored noises including white noise ($\beta = 0$), as shown in Figure 2. Therefore, we concluded that temporally anti-correlated noise improves exploration in environments with Noisy TV. In contrast, temporally correlated noise is relatively vulnerable to Noisy TV compared with temporally anti-correlated noise but improves exploration in overall environments.

### 5.3 EXPERIMENTS ON MINIGRID ENVIRONMENTS

To prove the effectiveness of the proposed TeCLE, we compared the experimental results with the baseline PPO, ICM, and RND in the Minigrid with and without Noisy TV. Considering notable outputs in Table 1 and Figure 2, we adopted red ($\beta = 2.0$) and blue ($\beta = -1.0$) noises as the default colored noise for TeCLE. The policy network is updated every 128 steps.

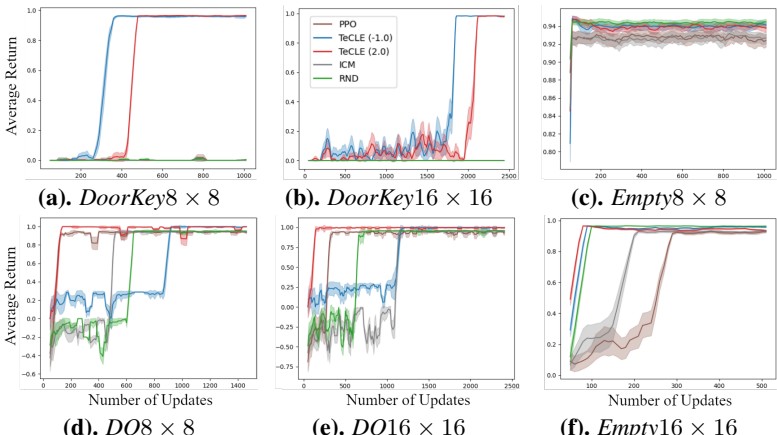

**Figure 4:** Comparison on Minigrid environments with Noisy TV. In *DoorKey*, other methods except for TeCLE failed to avoid the Noisy TV. Generally, red noise ($\beta = 2.0$) was more effective than other colored noises.

To demonstrate the exploratory behavior of TeCLE and compare its effectiveness with baselines, we measured the number of state visits by the agent (i.e., state coverage) (Raileanu & Rocktäschel, 2020; Kim et al., 2023). State coverage was measured by clipping when visitation exceeded 10k during training. It was then normalized to a range between 1 and 100. Figure 3 shows the state coverage in *DoorKey*16×16 and *Empty*16×16 environments. As shown in *DoorKey*16×16, whereas other baselines failed to open the door below and enter the other room, only TeCLE with red ($\beta = 2.0$) and blue ($\beta = -1.0$) noises succeeded in solving the tasks and learned the optimal policy for exploration. Additionally, it seems that blue noise ($\beta = -1.0$) encourages agents to exploit more than explore compared to red noise. In other words, red noise ($\beta = 2.0$) encourages agents to explore more than exploit compared to blue noise ($\beta = -1.0$). These exploratory behaviors depending on different colored noise also can be seen in *Empty*16 × 16. While TeCLE with red noise ($\beta = 2.0$) showed global exploration, blue noise ($\beta = -1.0$) showed local exploration. Moreover, the experimental results for all $\beta$ in Appendix D.2 show that as $\beta$ increases, TeCLE encourages the agent to explore more than exploit. This phenomenon is similar to the previous studies (Eberhard et al., 2023; Hollenstein et al., 2024) that adjusted exploratory behaviors of agents by applying colored noise for action selection. Thus, we concluded that the amount of temporal correlation is closely related to the exploratory behaviors as well as robustness to the Noisy TV.

Figure 4 shows the experimental results in the Minigrid environments with Noisy TV. In *DoorKey*8× 8, it is shown that only TeCLE can effectively learn the environments where Noisy TV exists, whereas other methods failed. In particular, TeCLE with blue noise ($\beta = -1.0$) showed faster convergence than the red noise ($\beta = 2.0$) in both *DoorKey*8 × 8 and *DoorKey*16 × 16 environments. This means that the improved exploitation from the temporal anti-correlation could be suitable for sparse reward environments with Noisy TV. On the other hand, in *DynamicObstacles* (denoted as *DO)*, TeCLE with red noise ($\beta = 2.0$) showed the faster convergence. As in *DoorKey* environments, the other methods failed to learn the optimal policy and avoid being trapped by Noisy TV. Notably, although the convergence of the TeCLE with blue noise ($\beta = -1.0$) was slightly slower than red noise ($\beta = 2.0$) due to the improved exploitation, it eventually converged to the highest average return. Furthermore, it seems that in easy environments such as *Empty*8 × 8, all methods converged to a high average return. However, in difficult environments such as *Empty*16 × 16, the convergence was slow for all methods except TeCLE. This is because the rewards become sparse as the state-space expands, and agents using other methods tend to lose curiosity about the environment.

Figure 5 shows the experimental results in the Minigrid environments without Noisy TV. We found that TeCLE with red ($\beta = 2.0$) and blue ($\beta = -1.0$) noises outperformed the baselines in overall environments. In *DoorKey*8×8, only the proposed TeCLE and RND seem to learn the optimal policy to solve the tasks. Besides, RND produced fast convergence in *DynamicObstacles*8 × 8. However, it is noted that RND converged to an average return of around 0.9, while TeCLE with red ($\beta = 2.0$) and blue ($\beta = -1.0$) noises converged around 1. In *DO*8×8, although PPO converged faster than TeCLE with blue ($\beta = -1.0$) noise, it converged slowly or even could not learn the policy and environments at all of the environments except *DO*8 × 8. The convergence of TeCLE with red ($\beta = 2.0$) and

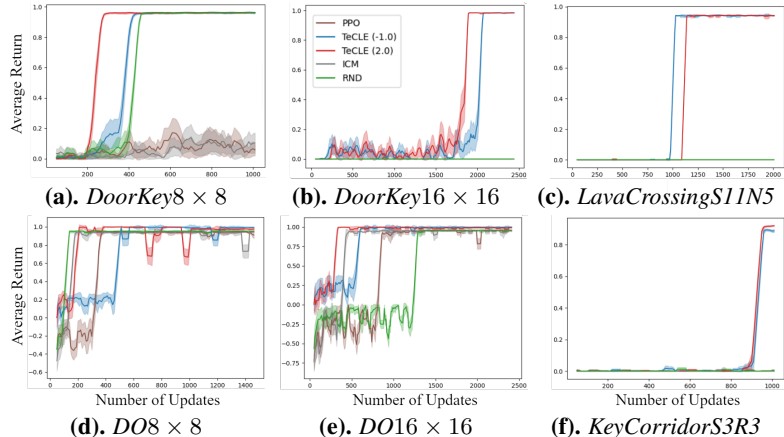

**Figure 5:** Comparison on Minigrid environments without Noisy TV. Only TeCLE can show convergence in both *LavaCrossingS11N5* (large state-space) and *KeyCorridorS3R3* (hard task).

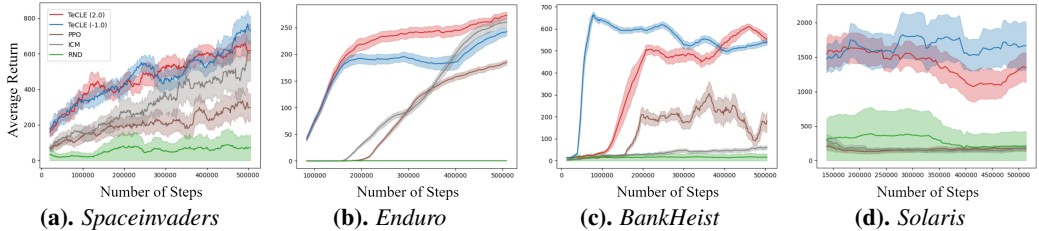

**Figure 6:** Comparison on Stochastic Atari environments. In the above hard and sparse reward environment, TeCLE outperformed other baselines, showing no significant difference between TeCLE with red ($\beta = 2.0$) and blue ($\beta = -1.0$) noises. Only TeCLE learned the environments while avoiding being trapped by stochasticity.

blue ($\beta = -1.0$) noises in *DoorKey*$16 \times 16$ demonstrates that the red noise ($\beta = 2.0$) would be more effective in learning the policy and environments if Noisy TV does not exist. Furthermore, it is noted that only TeCLE can show convergence in both *LavaCrossingS11N5* with large state-space and *KeyCorridorS3R3* with hard tasks.

## 5.4 EXPERIMENTS ON STOCHASTIC ATARI ENVIRONMENTS

To further investigate whether TeCLE can be robust to stochasticity or not, we evaluated it in the Stochastic Atari environments (Burda et al., 2018b; Pathak et al., 2019) and compared it with other baselines. As in the previous Minigrid experiments, we adopted red ($\beta = 2.0$) and blue ($\beta = -1.0$) noises as the colored noise for TeCLE.

Figure 6 shows the experimental results in several Stochastic Atari environments. Whereas *SpaceInvaders* and *Enduro* are known as easy and dense reward environments, *BankHeist* and *Solaris* are known as hard and sparse reward environments (Ostrovski et al., 2017). In *SpaceInvaders* and *Enduro*, TeCLE outperformed other baselines, showing no significant difference between red ($\beta = 2.0$) and blue ($\beta = -1.0$) noises. It also showed that all methods except for RND can handle stochasticity in dense reward environments. However, whereas other baselines failed to learn the *BankHeist* and *Solaris*, only TeCLE learned the environments while avoiding being trapped by stochasticity. Therefore, experimental results of the Stochastic Atari environments confirmed that the proposed TeCLE is the most effective in handling stochasticity in both dense and sparse reward environments. Notably, since blue noise ($\beta = -1.0$) performs better than red noise ($\beta = 2.0$) in the three environments, it can be concluded that the temporal anti-correlation makes the agents robust not only to Noisy TV but also to stochasticity.

## 5.5 ABLATION STUDY I: EFFECTS OF ACTION AS A CONDITION

To demonstrate the effects of action as a condition for the action-conditioned latent space of TeCLE, we experimented with an ablation study. Figure 7 shows the experimental results for analyzing the

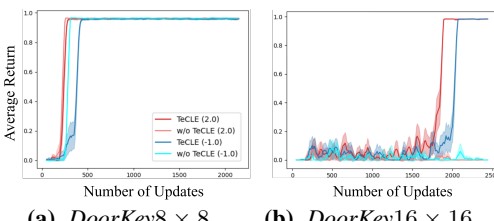

**(a).** *DoorKey* $8 \times 8$    **(b).** *DoorKey* $16 \times 16$

**Figure 7:** Comparison of effects of action as a condition in the Minigrid without Noisy TV. In the *DoorKey* $16 \times 16$, TeCLE without using action as a condition failed to learn the optimal policy.

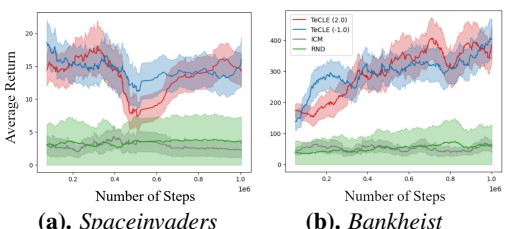

**(a).** *Spaceinvaders*    **(b).** *Bankheist*

**Figure 8:** Comparison of performance in the Stochastic Atari environments without extrinsic rewards. Results show that only TeCLE can learn about the environment without using extrinsic rewards.

effects of action as a condition. The term *TeCLE* refers to the TeCLE using action as a condition, while the term *w/o TeCLE* refers to the TeCLE without using the condition.

Since *DoorKey* $8 \times 8$ has a small state-space, the effects of action were not significant. However, in *DoorKey* $16 \times 16$, TeCLE without using action as a condition failed to learn the optimal policy. Therefore, it seems that the effects of the action were significant in terms of self-supervised learning. Also, it is shown that when an environment has a large state-space, the action-conditioned latent space can make better state reconstructions, helping find the optimal policy.

### 5.6 ABLATION STUDY II: EXPLORATION WITHOUT EXTRINSIC REWARD

To prove whether TeCLE can be robust in the absence of any extrinsic rewards, we additionally experimented with an ablation study. For experiments, we set the coefficient of extrinsic reward to zero and compared the average return of TeCLE with the baselines. Note that only intrinsic rewards are used to update the policy network of agents. Thus, extrinsic rewards are not used except for performance measurements. Since PPO does not use intrinsic rewards, it was not compared.

Figure 8 shows the experimental results in the Stochastic Atari environments when extrinsic rewards were absent, demonstrating that only TeCLE can learn the environments. Experiments were conducted on *SpaceInvaders* and *BankHeist*, which are dense and sparse reward environments, respectively. However, since the agent does not receive any extrinsic rewards, it was expected that the agent could not learn the environment. Surprisingly, the experimental results of both environments show that TeCLE can learn the environments without using extrinsic rewards. Most of all, TeCLE in *BankHeist* shows a similar average return to when extrinsic rewards are present, as shown in Figure 6 (c). Although RND is known to perform well in sparse reward environments and hard exploration tasks, the above experimental results show that TeCLE outperformed RND. In conclusion, the above ablation study shows that the effects of the intrinsic reward from the proposed TeCLE were considerable in the absence of extrinsic reward. Therefore, we expect that the proposed TeCLE can be more effective than other methods in real-world scenarios where rewards are sparse or absence.

## 6 CONCLUSION AND FUTURE WORK

This paper proposes TeCLE, representing a novel curiosity-driven exploration method that defines intrinsic rewards through states reconstructed from an action-conditioned latent space. Extensive experiments on benchmark environments show that the proposed method outperforms popular exploration methods such as ICM and RND and avoids being trapped by Noisy TV and stochasticity in the environments. Most of all, we find that the amount of temporal correlation is closely related to the exploratory behaviors of agents. Therefore, we recommend that the blue and red noise, which show notable performance among various colored noises, be the default settings for TeCLE in environments where noise sources exist and rewards are sparse, respectively. As far as we know, our study is the first approach to introduce temporal correlation and temporal anti-correlation to intrinsic motivation. Therefore, future studies are needed to verify that temporal correlation is effective in various intrinsic motivation methods, such as count-based exploration methods.

## ETHIC STATEMENT

This paper proposes a new intrinsic formulation for deep reinforcement learning agents. In this study, we have kept the following ethical principles of ICLR 2025:

1. **Contribute to Society and to Human Well-being:** This paper aims to enable agents in deep reinforcement learning to be robust to stochasticity, and to learn optimal policies in sparse reward environments through exploration. The proposed algorithm has potential applications in industries such as gaming, robot control, and autonomous driving.

2. **Uphold High Standards of Scientific Excellence:** We have intensively performed experiments to validate the proposed method. The motivations, ideas, and conclusions are presented to contribute to the scientific community.

3. **Avoid Harm:** The study does not include any human subjects or sensitive personal data. We strongly discourage any misuse of our work that could harm individuals, although there is no explicit information about the misuse in the manuscript.

4. **Be Honest, Trustworthy, and Transparent:** We have honestly reported our research findings, including both strengths and limitations. All data sources, model structure, and experimental environments are fully disclosed to ensure transparency.

5. **Be Fair and Take Action to Avoid Discrimination:** Because we adopt public experimental environments such as Atari and Minigrid using the Pytorch library, the experiments can be fair without any discrimination.

6. **Respect the Work Required to Produce New Ideas and Artefacts:** We cite all relevant references in the manuscript to respect existing works. This paper is written considering the previous works and knowledge.

7. **Respect Privacy:** The environments used in our experiments, such as Atari and Minigrid, are publicly available and do not contain personal information.

8. **Honour Confidentiality:** This paper does not have any confidentiality agreements.

## REPRODUCIBILITY STATEMENT

We adopt the Atari and Minigrid environments, which are public experimental environments for easy reproduction. The attached code as supplementary materials can be easily performed when the corresponding environments are ready. The hyperparameters for all experimental environments are well described in the Appendix.

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

## A PRELIMINARIES

### A.1 OPTIMIZATION OF CVAE

The goal of a VAE is to output $\hat{x}$ that has a similar distribution to the input data $x$. The VAE consists of an encoder $q_\lambda$ and a decoder $p_\psi$, where $q_\lambda$ encodes $x$ into the latent space $z$, and $p_\psi$ reconstructs the $\hat{x}$ from the $z$. In a dataset $X = \{x_1, ..., x_N\}$ that consists of $N$ independent and identically distributed (i.i.d.) samples, let us assume that each data $x \in X$ is reconstructed from $z$. For the optimization, VAE performs a density estimation on $P(x, z)$ to maximize the likelihood of the training data $x \in X$ formulated as:

$$\log P(x) = \sum_{i=1}^{N} \log P(x_i). \tag{9}$$

Since it is difficult to access marginal likelihood directly (Kingma, 2013), the parametric inference model $q_\lambda(z|x)$ is used to optimize a variational lower bound on the marginal log-likelihood as:

$$L_{\lambda,\psi} = E_{P(z|x)}[\log q_\lambda(x|z)] - KL(q_\lambda(z|x)||p_\psi(z))). \tag{10}$$

Then, the VAE reparameterizes $q_\lambda(z|x)$ to optimize the lower bound (Kingma, 2013; Rezende et al., 2014). In Eq.(10), the first term $E_{P(z|x)}[\log q_\lambda(x|z)]$ denotes the reconstruction loss of $\hat{x}$ from $z$, where the expectation is taken over the approximate posterior distribution $q_\lambda(z|x)$. The second term $KL(q_\lambda(z|x)||p_\psi(z)))$ denotes the KL divergence between the $q_\lambda(z|x)$ and the prior distribution $p_\psi(z)$ to regularize the distribution of latent space.

Our intrinsic formulation is based on the CVAE proposed by Sohn et al. (2015). The difference between CVAE and VAE is the use of a condition variable. Also, we adopt state $s$ as the input variable and action $a$ as the condition variable. Thus, Eq.(10) can be rewritten for the optimization of the proposed method as:

$$L_{\lambda,\psi} = E_{P(z|s,a)}[\log q_\lambda(a|s, z)] - KL(q_\lambda(z|s, a)||p_\psi(z|s)). \tag{11}$$

## A.2 PROPERTIES OF COLORED NOISE

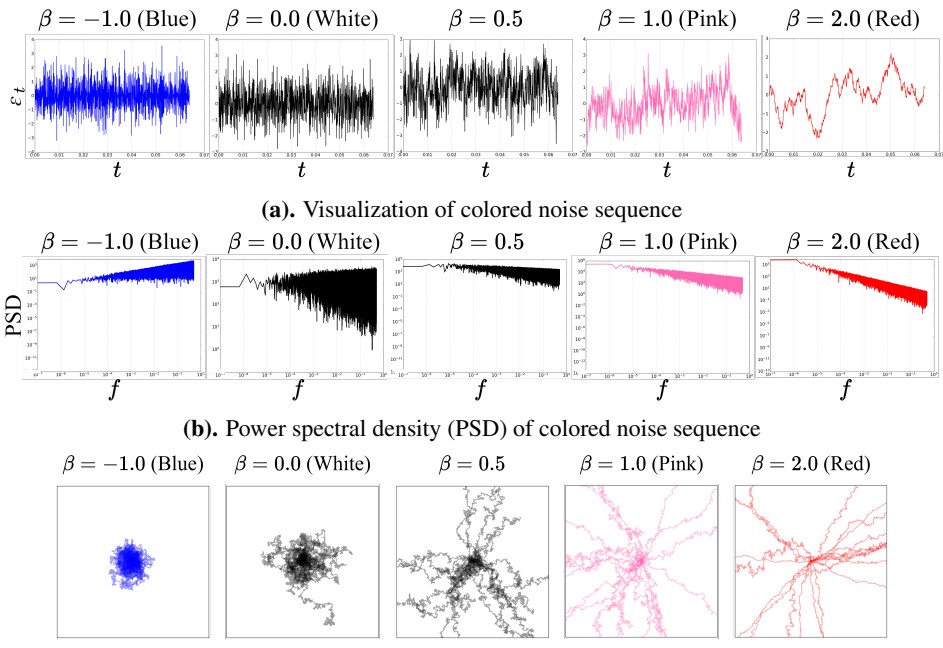

**(a).** Visualization of colored noise sequence

**(b).** Power spectral density (PSD) of colored noise sequence

**(c).** The trajectory of a two-dimensional random walk

**Figure 9:** Properties of colored noise depending on $\beta$.

Figure 9 (a) shows the visualization of the generated colored noise sequence $\varepsilon_{(1:t)}$ with length $t = 1000$. The noise sequence with low $\beta$ ($\beta < 0$) shows consistently large perturbations, while high $\beta$ ($\beta > 0$) shows generally small perturbations. As shown in Figure 9 (b), when observing the PSD in the frequency domain, the effects of various $\beta$ in colored noise sequences are visualized more clearly. Figure 9 (b) shows that the PSD of a colored noise sequence with low $\beta$ has more energy in the high frequency range, while high $\beta$ shows the opposite characteristics. Although the PSD of white noise ($\beta = 0$) in Figure 9 (b) show large fluctuations in the high frequency range, the average PSD of white noise can remain consistent across all frequency ranges. Therefore, it is concluded that the colored noise sequence with $\beta = 0$ (white noise) is temporally uncorrelated.

On the other hand, Figure 9 (c) shows two-dimensional random walks of different colored noises. It is shown that the random walk of a colored noise with low $\beta$ stays within a local range, which indicates that its motion is changed more frequently than those with higher $\beta$. Figure 9 (c) illustrates that the movement patterns in random walks are influenced by the $\beta$ of the colored noise. As $\beta$ increased, the area of random walks tended to become more extended. Since colored noise with a higher $\beta$ has more energy in the low-frequency range, the action frequency decreases in random walks.

## B  PSEUDO-CODE OF TECLE

Algorithm 1 shows the pseudo-code of the proposed TeCLE. We adopt PPO (Schulman et al., 2017) as a baseline RL algorithm. When training, the policy network of PPO is updated by using the combined values of intrinsic rewards from TeCLE and extrinsic rewards from the environments. Besides, we generated colored noise sequence using the *colorednoise* Python package[1], based on the procedure described by Timmer & Koenig (1995). The detailed operations of TeCLE are described in the supplementary materials.

---

**Algorithm 1** Temporally Correlated Latent Exploration

---

$N :=$ Number of rollouts,
$N_{update} :=$ Number of update steps,
$K :=$ Length of a rollout,
$B_i :=$ Batch in $i$-th rollout,
$R_i^I :=$ Intrinsic return in $i$-th rollout,
$A_i^I :=$ Intrinsic advantage in $i$-th rollout,
$R_i^E :=$ Extrinsic return in $i$-th rollout,
$A_i^E :=$ Extrinsic advantage in $i$-th rollout,
$\beta :=$ Color parameter,
$f_\theta :=$ Embedding network, $g_\theta :=$ Inverse network, $q_\lambda :=$ Encoder, $p_\psi :=$ Decoder,
$L_{recon} :=$ Reconstruction loss, $L_{KL} :=$ KL divergence loss, $L_{PPO} :=$ PPO loss,
$L_I :=$ Inverse loss

$t \leftarrow 1$
$s_1 \sim p(\emptyset)$             $\triangleright$ Transit to the initial state
**for** $i = 1$ **to** $N$ **do**
     $\varepsilon_{(1:K)} \leftarrow$ Noise_Sequence$(K, \beta)$     $\triangleright$ Generate $K$ values of colored noise with $\beta$ in advance
     **for** $j = 1$ **to** $K$ **do**
         $a_t \sim \pi(a_t|s_t)$            $\triangleright$ Sample $a_t$ from policy network
         $s_{t+1}, r_t^e \sim p(s_{t+1}, r_t^e|s_t, a_t)$     $\triangleright$ Sample the next state and receive extrinsic reward
         $\phi(s_{t+1}) \sim f_\theta(s_{t+1})$     $\triangleright$ Output next state representation from embedding network $f_\theta$
         $\hat{\phi}(s_{t+1}) \sim p_\psi(q_\lambda(\phi(s_{t+1}), a_t), a_t)$     $\triangleright$ Reconstruct $\hat{\phi}(s_{t+1})$ using colored noise $\varepsilon_{t+1}$
         $r_t^i \leftarrow \|\hat{\phi}(s_{t+1}) - \phi(s_{t+1})\|_2$         $\triangleright$ Compute intrinsic reward
         $B_i \leftarrow \{s_t, s_{t+1}, a_t, r_t^e, r_t^i, \hat{\phi}(s_{t+1}), \phi(s_{t+1})\} \cup B_i$     $\triangleright$ Include values in batch $B_i$
         $t \leftarrow t + 1$
     **end for**
     $r_i^i \leftarrow$ Normalize$(B_i)$         $\triangleright$ Normalize the intrinsic rewards in $B_i$
     $A_i^I \leftarrow$ Intrinsic_Advantage_Return$(B_i)$     $\triangleright$ Compute advantage for intrinsic rewards
     $R_i^I \leftarrow$ Intrinsic_Return$(B_i)$         $\triangleright$ Compute intrinsic returns
     $A_i^E \leftarrow$ Extrinsic_Advantage_Return$(B_i)$     $\triangleright$ Compute advantage for extrinsic rewards
     $R_i^E \leftarrow$ Extrinsic_Return$(B_i)$         $\triangleright$ Compute extrinsic returns
     $A_i \leftarrow A_i^I + A_i^E$         $\triangleright$ Compute combined advantages
     $R_i \leftarrow R_i^I + R_i^E$         $\triangleright$ Compute combined returns
     **for** $j = 1$ **to** $N_{update}$ **do**
         $\pi \leftarrow$ Update$(\pi, L_{PPO}(B_i, R_i, A_i))$     $\triangleright$ Update policy network w.r.t. $L_{PPO}$
         $f_\theta, g_\theta \leftarrow$ Update$(f_\theta, g_\theta, L_I(B_i))$     $\triangleright$ Update embedding and inverse network w.r.t. $L_I$
         $q_\lambda, p_\psi \leftarrow$ Update$(q_\lambda, p_\psi, L_{recon,KL}(B_i))$     $\triangleright$ Update CVAE w.r.t. $L_{recon}$ and $L_{KL}$
     **end for**
**end for**

---

---

[1] https://github.com/felixpatzelt/colorednoise

# C  IMPLEMENTATION DETAILS

## C.1  ENVIRONMENTS

In the experiments, we adopt widely used benchmark Minigrid environments developed by Chevalier-Boisvert et al. (2018). Besides, the Atari Learning Environment (ALE) (Bellemare et al., 2013), which is another widely used Atari benchmark, was adopted for the Stochastic Atari experiments. Tables 2 and 3 list the names and Gym Spec-ids of the experimented environments chosen among the Minigrid and Stochastic Atari environments.

**Table 2:** Names and Gym Spec-ids of experimented environments chosen among the Minigrid environments.

| Environment | Gym Spec-id |
|---|---|
| *Empty* $8 \times 8$ | MiniGrid-Empty-8x8-v0 |
| *Empty* $16 \times 16$ | MiniGrid-Empty-16x16-v0 |
| *DoorKey* $8 \times 8$ | MiniGrid-DoorKey-8x8-v0 |
| *DoorKey* $16 \times 16$ | MiniGrid-DoorKey-16x16-v0 |
| *KeyCorridorS3R3* | MiniGrid-KeyCorridorS3R3-v0 |
| *LavaCrossingS9N3* | MiniGrid-LavaCrossingS9N3-v0 |
| *LavaCrossingS11N5* | MiniGrid-LavaCrossingS11N5-v0 |
| *MultiRoomN2S4* | MiniGrid-MultiRoom-N2-S4-v0 |

**Table 3:** Names and Gym Spec-ids of experimented environments chosen among the Atari environments.

| Environment | Gym Spec-id |
|---|---|
| *Alien* | AlienNoFrameskip-v4 |
| *BankHeist* | BankHeistNoFrameskip-v4 |
| *Enduro* | EnduroNoFrameskip-v4 |
| *Montezuma's Revenge* | MontezumaRevengeNoFrameskip-v4 |
| *MsPacman* | MsPacmanNoFrameskip-v4 |
| *Qbert* | QbertNoFrameskip-v4 |
| *Skiing* | SkiingNoFrameskip-v4 |
| *Solaris* | SolarisNoFrameskip-v4 |
| *SpaceInvaders* | SpaceInvadersNoFrameskip-v4 |
| *Zaxxon* | ZaxxonNoFrameskip-v4 |

## C.2 PREPROCESSING

Table 4 shows the detailed information on preprocessing applied to all experiments. We adopted sticky actions (Machado et al., 2018) to introduce non-determinism in the environment, thereby preventing the memorization of action sequences. We also conducted experiments with three seeds for reproducibility.

Table 4: Details of preprocessing applied in all experiments.

| Hyperparameter | Value |
|---|---|
| Gray-scaling | True |
| Observation downsampling | $84 \times 84$ |
| Observation normalization | $x \mapsto x/255$ |
| Frame stack | 4 |
| Max and skip frames | 4 |
| Max frames per episode | 18K |
| Sticky action probability | 0.25 |
| Terminal on life loss | True |
| Seed | $\{1, 3, 5\}$ |
| Clip reward | True |
| Channel first | True |

## C.3 HYPERPARAMETERS

Table 5 shows the hyperparameters used for all experiments. Additional hyperparameters used in TeCLE are described in the supplementary material.

Table 5: Hyperparameters for Minigrid and Atari environments.

| Hyperparameter | Minigrid | Atari |
|---|---|---|
| Unroll length | 128 | 128 |
| Entropy coefficient | 0.01 | 0.001 |
| Value loss coefficient | 0.5 | 0.5 |
| Number of parallel environments | 16 | 32 |
| Learning rate | 0.001 | 0.0001 |
| Optimization algorithm | Adam | Adam |
| Batch size | 256 | 512 |
| Number of optimization epoch | 4 | 4 |
| Policy architecture | CNN | CNN |
| Policy gradient clip range | [0.8, 1.2] | [0.9, 1.1] |
| Coefficient of intrinsic reward | 0.99 | 0.99 |
| Coefficient of extrinsic reward | 0.99 | 0.999 |
| GAE $\lambda$ | 0.95 | 0.95 |
| Update every $N$ steps | 128 | 512 |

## C.4 NEURAL NETWORK ARCHITECTURES

**Table 6:** Neural network architecture of policy network and TeCLE for Atari environments.

| Part | Architecture |
|---|---|
| Policy Network | 3 convolutional layers ([32, 64, 64] output channels, $8 \times 8, 4 \times 4, 3 \times 3$] kernel size, [4, 2, 1] stride, 0 padding), hidden ReLU layer, 2 MLP layers (256, 448) dimension, followed by 2 value heads (intrinsic value, extrinsic value). |
| TeCLE | **Embedding Network**: 4 convolutional layers ([32, 32, 32, 32] output channels, $3 \times 3$ kernel size, 2 stride, 1 padding), hidden ReLU layer. 

 **Inverse Network**: 2 MLP layers (256, action dimension) output dimensions, hidden ReLU layer. 

 **Encoder**: 3 convolutional layers ([32, 32, 64] output channels, $1 \times 1$ kernel size, 1 stride, 0 padding), hidden ReLU layer, 2 MLP layers (256, 128) output dimensions, followed by 2 heads (mean, variance). 

 **Decoder**: 4 MLP layers ([64, 128, 256, state shape]) output dimensions, hidden Sigmoid layer. |

Table 6 shows the neural network architecture of the policy network and TeCLE used for the Atari environments. Our policy network has two value heads (intrinsic and extrinsic values). The overall architecture of TeCLE in Figure 1, consists of the embedding network, inverse network, encoder, and decoder. On the other hand, in the Minigrid environments, the convolutional layer of the policy network and embedding network are adjusted to 3 convolutional layers ([16, 32, 64] channels, $2 \times 2$ kernel size, 1 stride, 0 padding) and 3 convolutional layers ([32, 32, 32] channels, $3 \times 3$ kernel size, 2 stride, 1 padding), respectively.

# D    EXPERIMENTS OF TECLE WITH VARIOUS COLORED NOISE

To further investigate the effects and differences of colored noises, we experimented TeCLE with various color noise $\beta \in \{-1.0, 0.0, 0.5, 1.0, 2.0\}$. It is notable that various colored noises corresponding to different $\beta$ not only has a significant impact on the performance of the agent but also affects the exploratory behavior.

## D.1    EXPERIMENTS ON MINIGRID ENVIRONMENTS

Figure 10 shows the experimental results of TeCLE with various $\beta$ in the Minigrid environments without Noisy TV. The overall experimental results show that temporally correlated noise and anti-correlated noise ($\beta \neq 0$) perform better than temporal uncorrelated noise ($\beta = 0$) for TeCLE. Besides, as shown in *DoorKey*$16 \times 16$, *LavaCrossingS11N5*, and *KeyCorridorS3R3* environments, it is notable that $\beta$ determines exploratory behavior, varying the performance of the agent. This demonstrates not only that the amount of temporal correlation has a significant impact on the exploration of the agent, but also provides a reason for the difference in performance compared to baselines PPO, ICM, and RND, as shown in Section 5.

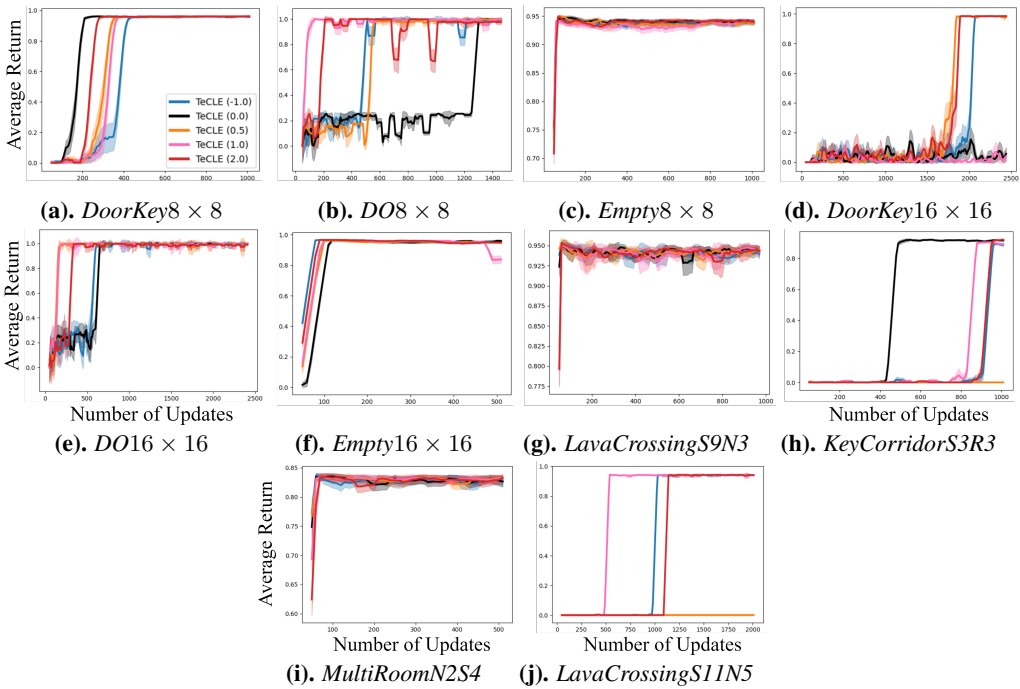

**Figure 10:** Comparison on Minigrid environments without Noisy TV.

On the other hand, Figure 11 shows the experimental results of TeCLE with various $\beta$ in the Minigrid environments with Noisy TV. Whereas red noise ($\beta = 2.0$) generally shows a better performance than other noises in the above experiments due to the improved exploration, blue noise ($\beta = -1.0$) outperforms other baselines in *DoorKey* environments due to the improved exploitation and robustness to Noisy TV. As shown in *DynamicObstacles*, although the improved exploitation of blue noise ($\beta = -1.0$) leads to a slightly slower convergence compared to other noises, it significantly outperforms the baselines, as shown in Figures 11 (b) and (e).

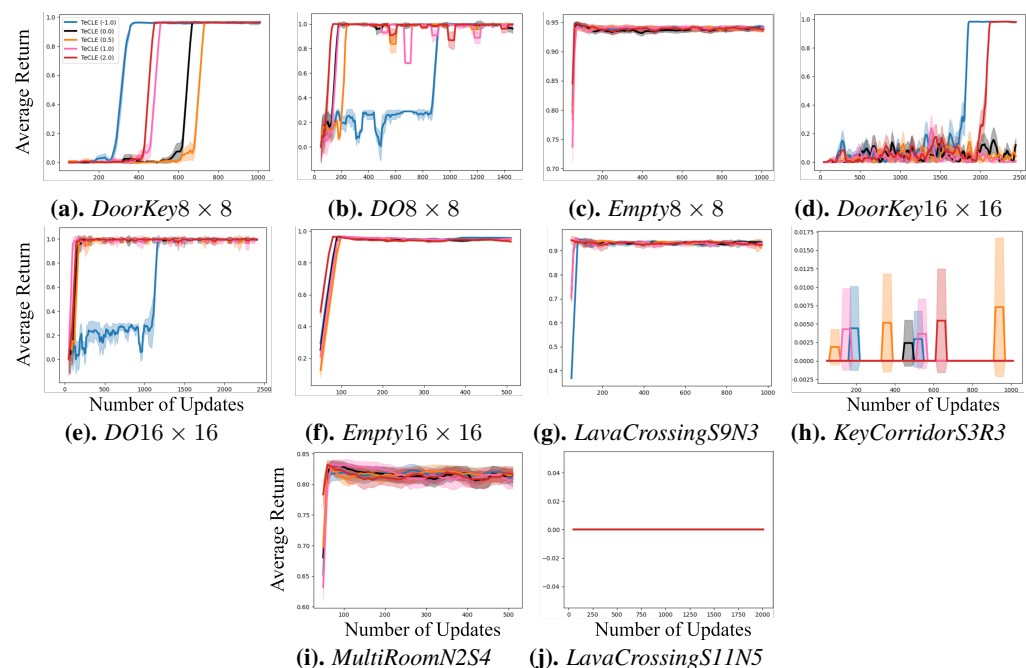

**(a).** *DoorKey*8 × 8  **(b).** *DO*8 × 8  **(c).** *Empty*8 × 8  **(d).** *DoorKey*16 × 16

**(e).** *DO*16 × 16  **(f).** *Empty*16 × 16  **(g).** *LavaCrossingS9N3*  **(h).** *KeyCorridorS3R3*

**(i).** *MultiRoomN2S4*  **(j).** *LavaCrossingS11N5*

**Figure 11:** Comparison on Minigrid environments with Noisy TV.

## D.2 STATE COVERAGE ON MINIGRID ENVIRONMENTS

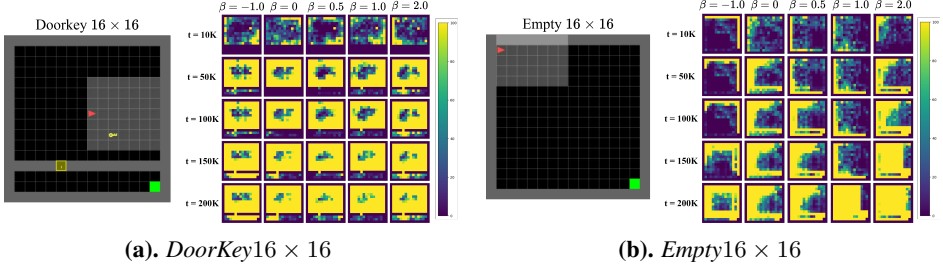

**(a).** *DoorKey*16 × 16  **(b).** *Empty*16 × 16

**Figure 12:** Visualized state coverage in *DoorKey*16 × 16 and *Empty*16 × 16 without Noisy TV of TeCLE with various $\beta$. The above visualization shows that TeCLE encourages agents to explore as $\beta$ increases. Notably, in *Empty*16 × 16, several results of TeCLE with temporally correlated noise ($\beta > 0$) tends to explore rather than exploit. In other words, TeCLE with temporally anti-correlated noise ($\beta < 0$) tends to exploit rather than explore. On the other hand, temporally uncorrelated noise ($\beta = 0$) shows the intermediate degree between exploration of $\beta = 2.0$ and exploitation of $\beta = -1.0$.

To demonstrate the exploratory behaviors of TeCLE with various colored noises, we measured the state coverage in the Minigrid environments, as shown in Figure 12. We counted the states visited by the agent for a total of 200k frames during training. Besides, the state coverage was measured by clipping when visitation exceeded 10k. It was then normalized to a range between 1 and 100 and represented as a heatmap. Interestingly, as $\beta$ in colored noise increased, it seems that TeCLE encourages agents to explore rather than exploit. Therefore, TeCLE with temporally correlated noise ($\beta > 0$) tends to explore globally, while temporally anti-correlated noise ($\beta < 0$) explores locally. As demonstrated in Section A.2, the reason is that the smooth changing magnitude of the noise sequence of temporally correlated noise ($\beta > 0$) allows agents to assign large intrinsic rewards to novel states. Therefore, it is concluded that the agent of TeCLE with red noise ($\beta = 2.0$) continues to explore until the end of the training, achieving high state coverage. On the other hand, the constantly fluctuating magnitude of temporally anti-correlated noise ($\beta < 0$) allows agents to assign smaller intrinsic rewards, leading to exploitation rather than exploration. In other words,

fluctuating intrinsic reward of temporally anti-correlated noise ($\beta < 0$) makes agents less sensitive to novel states. Therefore, as we concluded in Section 5, the state coverage in Figure 12 shows that the amount of temporal correlation is closely related to the exploratory behaviors of agents.

Furthermore, the different exploratory behaviors of TeCLE with various $\beta$ suggest that our approaches can outperform existing curiosity-based methods such as ICM and RND, which maintain their exploratory behavior even when the characteristics of the environment change.

### D.3 EXPERIMENTS ON STOCHASTIC ATARI ENVIRONMENTS

Figure 13 shows the experimental results of TeCLE with various $\beta$ in the Stochastic Atari environments. Similar to the experimental results on the Minigrid environments in Appendices D.1 and D.2, the overall experimental results showed that TeCLE with temporally correlated and anti-correlate noises ($\beta \neq 0$) outperformed the case with white noise ($\beta = 0$). Furthermore, in *Enduro* environments, agents with colored noises except for red ($\beta = 2.0$), blue ($\beta = -1.0$), and white ($\beta = 0$) noises were unable to learn the policy since they became trapped by stochasticity in the environment. On the other hand, pink noise ($\beta = 1.0$) showed better performance than other colored noises in *Solaris*. However, compared to red ($\beta = 2.0$) and blue ($\beta = -1.0$) noises, pink noise ($\beta = 1.0$) showed degraded performance in other experiments.

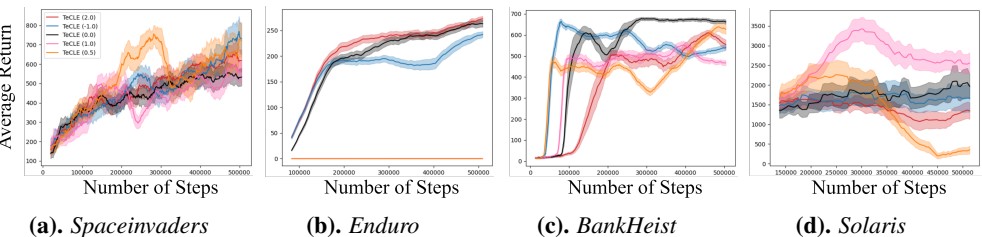

**(a).** *Spaceinvaders*  **(b).** *Enduro*  **(c).** *BankHeist*  **(d).** *Solaris*

**Figure 13:** Experimental results of TeCLE with various $\beta$ on Stochastic Atari environments.

# E    HARD EXPLORATION TASKS IN STOCHASTIC ATARI ENVIRONMENTS

To prove the exploration ability of curiosity agents, successfully exploring hard exploration environments is as important as successfully exploring environments while avoiding being trapped by noise sources. Thus, we conducted experiments on several hard exploration tasks (Bellemare et al., 2016) in Stochastic Atari environments and compared TeCLE with baselines. Considering the notable performance, red ($\beta = 2.0$) and blue ($\beta = -1.0$) noises were adopted as default colored noises for TeCLE.

As shown in Figure 14, although our proposed TeCLE aims to enhance exploration in environments where noise sources exist, it showed better performance in overall hard exploration tasks than baselines PPO, ICM, and RND. Whereas ICM and RND outperformed TeCLE in *Skiing* and *Zaxxon*, TeCLE outperformed them in the rest of the environments. It is notable that RND, which was proposed to enhance exploration in hard exploration tasks, performed worse than TeCLE in all environments except for Montezuma's Revenge. On the other hand, TeCLE with red ($\beta = 2.0$) and blue noise ($\beta = -1.0$) showed comparable performance across most environments, except for *Alien* and *Qbert*. As a result, we conclude that our proposed TeCLE can enhance the exploration ability of curiosity agents in hard exploration tasks while avoiding being trapped by stochasticity.

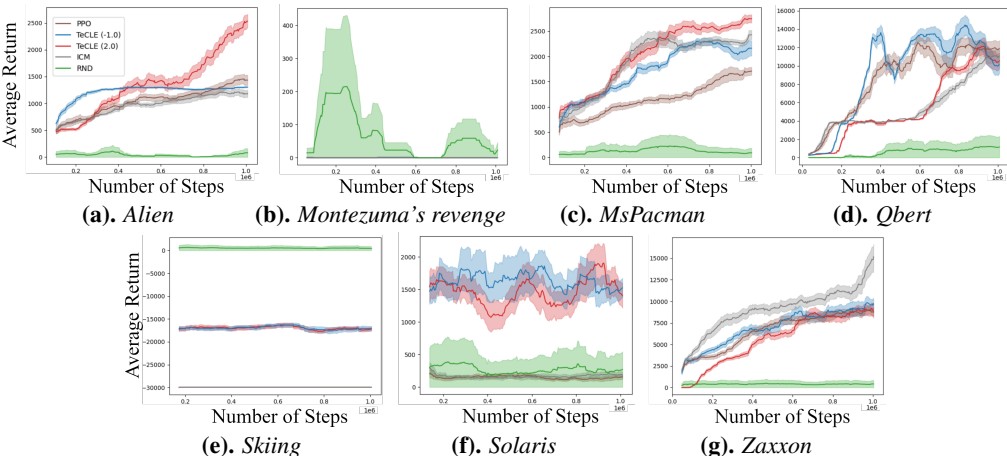

**Figure 14:** Comparison on hard exploration of Atari environments.

# F    COMPARISON OF INTRINSIC REWARDS

In this section, we compared the intrinsic rewards of TeCLE and those of baselines to explain how TeCLE can be robust to noise sources while outperforming baselines. Figure 15 shows that only TeCLE can learn the optimal policy network in Minigrid *DoorKey* $8 \times 8$ and $16 \times 16$ environments. The intrinsic rewards measured during training of the policy networks are shown in Figure 16. While the intrinsic reward of the baselines shows a small value near zero, the intrinsic reward of TeCLE maintains a relatively large value. As we hypothesized in Section 4, the reason for the difference in training and intrinsic reward between baselines and TeCLE is that CVAE in the TeCLE continuously injects noise when reconstructing state representation. Therefore, unlike baselines that maintain smaller intrinsic rewards since they minimize the prediction error of the state representation, TeCLE maintains a large intrinsic reward since it contains noise regardless of whether it is sufficiently explored. As a result, this tendency of intrinsic reward from TeCLE helps agents prevent being trapped in environments that contain inherently unpredictable noise sources.

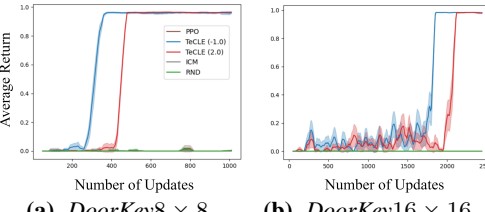

**(a).** *DoorKey*$8 \times 8$    **(b).** *DoorKey*$16 \times 16$

**Figure 15:** Comparison of average return on Minigrid environments with Noisy TV.

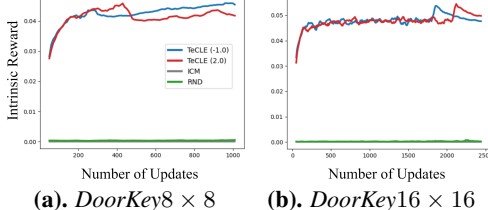

**(a).** *DoorKey*$8 \times 8$    **(b).** *DoorKey*$16 \times 16$

**Figure 16:** Comparison of intrinsic reward in Minigrid environments with Noisy TV.

