# OpenReview forum: "A Temporally Correlated Latent Exploration for Reinforcement Learning"
_ICLR.cc/2025/Conference — Submitted to ICLR 2025_

### Official Review · Reviewer_ttZZ · 2024-11-02

**Soundness:** 2
**Presentation:** 2
**Contribution:** 1
**Rating:** 3
**Confidence:** 5

**Summary:**

The paper proposes a method for improving exploration in RL using intrinsic motivation. Similar to ICM, a state embedding is learned with respect to an inverse dynamics loss and the error between a predicted embedding of the next state and the actual embedding of the next state serves an auxiliary reward. However, unlike ICM, the predicted embedding of the next state is obtained by a forward model but by reconstructing the next state embedding using a VAE. The paper proposes to use an action-conditioned C-VAE and to use colored noise while sampling the latent variable of the C-VAE to induce temporal correlation (or anti-correlation). The method is evaluated on suitable gridworlds (DoorKey) and Atari environments, where it achieved better results than the baselines (ICM and RND).

**Strengths:**

- The description of the approach is clear.
- While colored noise was previously used for sampling temporal correlated actions for better exploration, injecting temporal correlations while sampling the latent of the VAE is novel.

**Weaknesses:**

- The contributions are quite limited. The main contribution seems to be sampling the VAE latent with colored noise.
- The use of colored noise while sampling the VAE latent is not well motivated.
- Related work is insufficient discussed. For example, Kubovcik et. al used the reconstruction of an auto-encoder as intrinsic reward, Klissarov et al. also used a variational autoencoder, but used the KL between the posterior and prior for intrinsic reward, and, recently, Yan et al. also proposed to use the reconstruction error of a VAE for intrinsic reward but extended it with adaptive exploration mechanism, based on a discriminator.
- The empirical evaluation is insufficient. Table 1 does not show confidence intervals or baseline results. It is not clear how important hyperparameters (e.g. weight of the intrinsic reward) for the baseline were selected and which values were used, and, furthermore, important baselines are missing, e.g. the above mentioned related works.
- The writing could be improved and is often inaccurate. For example the abstract states that the latent space "models" the probability distribution of states, and that the environment where the agent performs best depends on the amount of temporal correlation. Some claims are also questionable, e.g. that the existing curiosity-driven methods are vulnerable to Noisy TV problems (some are, but not all of them)

References:

Martin Kubovcik, Iveta Dirgova Luptakova and Jiri Pospichal. Signal Novelty Detection as an Intrinsic Reward for Robotics. Sensors 2023.

Martin Klissarov, Riashat Islam, Khimya Khetarpal and Doina Precup. Variational State Encoding as Intrinsic Motivation in Reinforcement Learning. Task-Agnostic Reinforcement Learning Workshop at ICLR. 2019.

Renye Yan, Yaozhang Gan, You Wu, Ling Liang, Junliang Xing, Yimao Cai and Ru Huang. The Exploration-Exploitation Dilemma Revisited: An Entropy Perspective. ArXiV 2024.

**Questions:**

Could you extend on your motivation for colored noise? I understand that temporal correlation and anti-correlation will affect the smoothness of the intrinsic reward. But how does it relate to the noisy TV problem?

---

> ### Author Response · Authors · 2024-11-20
> **Rebuttal for Reviewer ttZZ**
>
> We thank the reviewers for their thorough review, which helped us improve the quality of this manuscript.
>
>
>
>
> ---
> * **W1:** The contributions are quite limited.
>
> * **A1:** Concerning W1, the reviewer pointed out that the contribution of the paper is relatively marginal. However, even if the contribution seems relatively marginal, **the proposed TeCLE is the first approach to using temporal correlation for curiosity-based intrinsic motivation.** We would also like to emphasize that the **TeCLE is well-behaved and shows outstanding performance compared to the baseline in a hard exploration task, a sparse reward environment, as well as an environment with noise sources.** We consider this a valuable contribution to the RL community.
>
>
>
>
> ---
> * **W2:** The use of colored noise while sampling the VAE latent is not well motivated.
>
> * **A2:** Concerning W2, inspired by the previous work [1, 2], which showed superior results by using colored noise as action noise, our work investigated the effect of colored noise for intrinsic reward by injecting it into the latent space. As shown in section 5, we found that applying colored noise to the latent space changed the exploratory behavior of the agents. **We thought that our motivation is similar to previous works in that we replace Gaussian noise with colored noise, but more advanced in that we extended it to curiosity-based intrinsic motivation method.**
>
>
>
>
> ---
> * **W3:** Related work is insufficient discussed.
>
> * **A3:** Concerning W3, we agree with the reviewer that the related works are not sufficiently discussed. Therefore, we have revised the section 2.3 in the paper as follows:
>
>
> >“2.3 CONDITIONAL VARIATIONAL AUTO-ENCODER (CVAE) FOR EXPLORATION
>
> >CVAE (Sohn et al., 2015) was introduced to learn the unlabeled dataset efficiently. It consists of an encoder and a decoder, which require condition variables and input variables. Since input variables are encoded as probability distributions into the conditioned-latent spaces, the policy of RL agents can be efficiently modeled. Thus, several studies adopted CVAE to mitigate the OOD problem in offline-RL. Zhou et al. (2021) employed CVAE to model the behavior policies of agents for a dataset or pre-collected experiences. The policy network was trained from the latent behavior space, and its decoder was used to output actions from the behavior space of the environment. Since the latent space after training was fit for the dataset distribution, the OOD problem of generating unpredictable actions could be mitigated. Besides, Rezaeifar et al. (2022) computed intrinsic reward for anti-exploration using the L2-norm between the predicted action by a decoder and actual action. Unlike previous studies (Klissarov et al., 2019; Kubovˇc´ık et al., 2023; Yan et al., 2024) that adopted VAE for intrinsic motivation, numerous studies adopted CVAE to model the policy networks.”
>
> ---
> * **W4:** The empirical evaluation is insufficient.
>
> * **A4:** Concerning W4, **we have added experiments with three additional Minigrid environments to Table 1:** $LavaCrossingS11N5$, $KeyCorridorS3R3$, and $MultiRoomN2S4$, and **modified Figure 2 in the paper.** The results of the another experiments in section 5 showed that temporally correlated/anti-correlated noise affects the exploration/exploitation performance of the agent. We also provided additional experiments on the $\beta$ of colored noise in Appendix D.
> &nbsp; &nbsp; &nbsp; &nbsp; In Figure 10, it can be seen that in environments with small state spaces, such as $Doorkey8\times8$ and $LavaCrossingS9N3$, all colored noise learned the optimal policy. However, as the state space becomes larger, such as $DoorKey16\times16$ and $LavaCrossingS11N5$, the difference in performance of the policy networks according to $\beta$ becomes obvious, and only some of the colored noise learns the optimal policy. **Even if empirical evaluation seems lacking, experimental results in section 5 and appendix indicates that the $\beta$ of the colored noise has a significant impact on the performance of the agent.**
> ---
> * **Q1:** Could you extend on your motivation for colored noise?
> * **A5:** For Q1, we explained the motivation for color noise in W2, please refer to it.
> ---
> **Reference**
>
>
>
>
>
> [1] Hollenstein, Jakob, Georg Martius, and Justus Piater. "Colored Noise in PPO: Improved Exploration and Performance Through Correlated Action Sampling." Proceedings of the AAAI Conference on Artificial Intelligence. Vol. 38. No. 11. 2024.
>
>
>
>
>
> [2] Eberhard, Onno, et al. "Pink noise is all you need: Colored noise exploration in deep reinforcement learning." The Eleventh International Conference on Learning Representations. 2023.

---

> > ### Comment · Reviewer_ttZZ · 2024-11-26
> >
> > Thank you for updating the related work.
> >
> > However, I argue my main concerns are still valid:
> >
> > - The use of colored noise is still not well-motivated. I do not think that it is sufficient to just say that you motivate it similar to colored noise on actions, because the effect on exploration is very different. Improving action correlation during exploration can be very sensible, in particular when jerky actions (as produced by Gaussians) perform bad. Applying colored noise while sampling from the VAE is conceptually very different and currently not well motivated.
> >
> > - Adding colored noise to the VAE is not a significant contribution. In particular, given that this modification is neither well-motivated nor produces very convincing benefits, the contribution seems too marginal.
> >
> > - The authors did not reply to my comment regarding the bad reproducibility due to missing details about hyperparameter search.

---

> ### Author Response · Authors · 2024-11-24
> **Reminder from Authors regarding the Discussion Period**
>
> Dear Reviewer,
>
> We would like to kindly remind you that we submitted our responses to your comments last Wednesday. As the discussion period is approaching its end, we sincerely ask for your attention to review our response. If you find that our answers sufficiently address your concerns and questions, we would be grateful if you could consider revising your rating.
>
> Thank you very much for your time and effort, and we truly appreciate your valuable feedback.

---

> ### Author Response · Authors · 2024-12-01
> **Answer for the first comment of reviewer ttZZ**
>
> Dear Reviewer ttZZ, we appreciate your insightful feedback and constructive suggestions.
>
> ---
> *  **W1:** The use of colored noise is still not well-motivated, and the contribution seems too marginal.
>
> *  **A1:** We sincerely appreciate the reviewer's thoughtful advice. We are sorry that you feel that applying colored noise to cVAE might appear to be a marginal contribution. However, we would like to confidently state that we made the following significant contributions:
>
> &nbsp;&nbsp; &nbsp;&nbsp;Our study focuses on proposing a novel curiosity-based intrinsic reward method that leverages cVAE to compute intrinsic rewards. This approach is notably robust against NoisyTV issues and actively promotes exploratory behavior. **Moreover, our findings reveal that incorporating colored noise into cVAE significantly influences the exploratory behavior of agents, as evidenced by extensive experiments conducted in both Minigrid and Atari environments.** Besides, in the revised version, we have demonstrated the effectiveness and contribution of the proposed method by providing not only many experiments in Appendices D and E but also additional experimental results, especially during this review period, as shown in **Appendix F and rebuttal (please refer to rebuttal for reviewer KD3W).** Notably, in Appendix F, **as shown in Figure 16, the measured intrinsic reward of the baselines remains close to zero, whereas the intrinsic reward of TeCLE consistently maintains a relatively large value.** This demonstrates that the proposed method introduces a critical difference compared to existing approaches.
>
> &nbsp;&nbsp; &nbsp;&nbsp;We believe these contributions and the demonstrated superior performance hold meaningful value for advancing the reinforcement learning field.
>
>   ---
>
> *  **W2:** The authors did not reply to my comment regarding the bad reproducibility due to missing details about the hyperparameter search.
>
> *  **A2:** We sincerely apologize for overlooking the reviewer's concern regarding the hyperparameter search in our initial response. **The details about the hyperparameters can be found in Appendix C.3**, while additional implementation details are provided in Section 5.1 and Appendix C. **Furthermore, to ensure reproducibility, we attached the implementation code for TeCLE in the supplementary materials.** We appreciate your attention to this matter and thank you for bringing it to our notice.
>
> ---
> Additionally, we have performed additional experiments to show the efficiency of the proposed TeCLE. Please consider them.
> Once again, we appreciate your response.  **If you and other reviewers have any further concerns during the remaining discussion period, please let us know, and we will do our best to address them.**  Thank you again.

---

### Official Review · Reviewer_GN7S · 2024-11-03

**Soundness:** 3
**Presentation:** 3
**Contribution:** 2
**Rating:** 6
**Confidence:** 4

**Summary:**

This paper proposes Temporally Correlated Latent Exploration (TeCLE), a novel curiosity-driven exploration method for reinforcement learning that uses action-conditioned latent spaces and temporal correlation, for addressing the noisy-TV issue, a long-standing problem in efficient exploration in RL. The authors propose to use action-conditioned latent space to model state distributions and compute intrinsic rewards based on reconstruction error. Temporal correlation is induced through colored noise in the latent sampling process, and different noise profiles lead to different exploratory behaviours. The method is evaluated on Minigrid and Stochastic Atari environments, showing improved robustness to the Noisy TV problem and stochasticity compared to baselines like ICM and RND.

**Strengths:**

- The proposed method is a novel combination of action-conditioned latent space and temporally correlated signals for addressing the noisy-TV problem.
- The empirical evaluations is comprehensive, with extensive experiments and ablation studies across multiple environments.
- The propose method empirically exhibits robustness with respect to noise.

**Weaknesses:**

- Empirical evaluations on hard exploration tasks (e.g., Montezuma's Revenge) show modest improvement.
- The theoretical grounding between the injected temporal correlations and resulting exploratory behaviour requires further justification.
- The proposed method induces significant computational overhead compared to simpler methods.

**Questions:**

See Weaknesses.

---

> ### Author Response · Authors · 2024-11-20
> **Rebuttal for Reviewer GN7S**
>
> We are indebted to the reviewers for their meticulous review and constructive advice.
>
> ---
> * **W1:** Empirical evaluations on hard exploration tasks (e.g., Montezuma's Revenge) show modest improvement.
>
> * **A1:** Concerning W1, we did not mentioned it in the paper because the main goal of our paper was to enable agents to learn in environments with noise sources (Noisy TV and stochasticity), not hard exploration tasks, and in fact, many studies that proposes intrinsic reward formulations similar to ours and whose goal is not hard exploration tasks rarely perform experiments on hard exploration tasks [1, 2]. **Nevertheless, our proposed TeCLE has been tested on hard exploration tasks (Figure 14) and has shown good performance. In addition, TeCLE showed better performance than RND, which is known to perform well on hard exploration tasks, as well as in environments with noise sources (Figures 4, 6, and 11).**
>
>
>
>
> ---
> * **W2:** The theoretical grounding between the injected temporal correlations and resulting exploratory behaviour requires further justification.
>
> * **A2:** Concerning W2, the discussion of temporal correlation and the exploratory behavior of agents was in the article on lines 253-267. Additionally, **we discussed the exploratory behavior of the agent according to $\beta$ of the various colored noise in Appendix D.2.**
>
> ---
> * **W3:**  The proposed method induces significant computational overhead compared to simpler methods.
>
> * **A3:** Concerning W3, we understand your concerns about the computation overhead of the proposed method. However, in the experiment in this paper, when we have measured the time for the Atari game on Nvidia RTX 4090, the RND was *6.422 learner steps/sec*, the ICM was *6.399 learner steps/sec*, and the TeCLE was *6.910 learner steps/sec*, all three methods took similar computation time. This is because TeCLE uses 4 frames for training the policy network, but only 1 frame is used for the intrinsic reward calculation, while ICM requires 4 frames for the policy network and intrinsic reward calculation. Besides, the architecture of CVAE is relatively simpler than the predictor network of RND, and we adopted CNN instead of MLP (the neural network architecture is specified in Appendix C.4). **This explains how TeCLE can have better performance even with similar computation time compared to the baselines.**
>
>
>
>
> ---
> **Reference**
>
>
>
>
>
> [1] Raileanu, Roberta, and Tim Rocktäschel. "Ride: Rewarding impact-driven exploration for procedurally-generated environments." arXiv preprint arXiv:2002.12292 (2020).
>
>
>
>
>
> [2] Mavor-Parker, Augustine, et al. "How to stay curious while avoiding noisy tvs using aleatoric uncertainty estimation." International Conference on Machine Learning. PMLR, 2022.

---

> ### Author Response · Authors · 2024-11-24
> **Reminder from Authors regarding the Discussion Period**
>
> Dear Reviewer,
>
> We would like to kindly remind you that we submitted our responses to your comments last Wednesday. As the discussion period is approaching its end, we sincerely ask for your attention to review our response. If you find that our answers sufficiently address your concerns and questions, we would be grateful if you could consider revising your rating.
>
> Thank you very much for your time and effort, and we truly appreciate your valuable feedback.

---

### Official Review · Reviewer_KD3W · 2024-11-04

**Soundness:** 1
**Presentation:** 1
**Contribution:** 1
**Rating:** 5
**Confidence:** 3

**Summary:**

This paper proposes to add temporal correlation to intrinsic rewards computation by learning an action conditional C-VAE for regularizing the representation of the state representation. The experiments in a few MiniGrid and Atari games show some improvements.

**Strengths:**

The connection between temporal correlation and the NoisyTV problem could be interesting.

**Weaknesses:**

## Writing
Overall, this paper only states how they implement the method but barely talks about the motivation behind them, which will make the reader lose. For example, in Line 48, the author is supposed to say how they address the noisy tv problem mentioned in the previous paragraph, but the author directly jumps in the implementation details of their method and never mentions why temporal correlation can mitigate NoisyTV problems. There are more examples like this, but it will take forever to enumerate all of them.

## Experiments
This paper lacks a lot of stronger baselines in the chosen environments (e.g., [1]). RND and ICM are not state-of-the-art in those environments. Moreover, only a fraction of Atari games are chosen, and the most classic hard-exploration problem, Montezuma Revenge, is not studied, which makes the results questionable.


[1] Zhang, Tianjun, Huazhe Xu, Xiaolong Wang, Yi Wu, Kurt Keutzer, Joseph E. Gonzalez, and Yuandong Tian. "Noveld: A simple yet effective exploration criterion." Advances in Neural Information Processing Systems 34 (2021): 25217-25230.

**Questions:**

N/A

---

> ### Author Response · Authors · 2024-11-20
> **Rebuttal for Reviewer KD3W**
>
> We sincerely thank the reviewers for their careful evaluation and valuable input.
>
>
>
>
> ---
> * **W1:** This paper only states how they implement the method but barely talks about the motivation behind them.
>
> * **A1:** Concerning W1, in the paper, we briefly described the formulation to explain how our method solves the noisy TV problem. We then explained it in detail in Line 55, **so it cannot be said that we did not mention it**. In addition, **we had motivation and theoretical explanations in Lines 250-264 of the paper, and experimental results in sections 5.2-5.4 described experimental results that support the ideas and hypotheses in the paper.**
> ---
> * **W2:** This paper lacks a lot of stronger baselines in the chosen environments (e.g., NovelD).
>
> * **A2:** Concerning W2, NovelD calculates intrinsic reward based on Novelty, which is measured using intrinsic motivation methods such as [1] and [2]. In this respect, unlike TeCLE, NovelD does not propose a new model or methodology, and while TeCLE may be applicable to NovelD, we do not consider it comparable. Also, our goal in this paper was to enable agents to learn in an environment with noise sources, and to the best of our knowledge, there is no curiosity-based intrinsic motivation method to address this using intrinsic rewards. Therefore, a comparison with baselines such as [1] and [3] was adopted. **However, as you pointed out, we have modified the statements of references to [1] and [3] as “state-of-the-art baselines” in the paper to “strong baselines.”**
> &nbsp; &nbsp; &nbsp; &nbsp; Also, as mentioned in Lines 274-275 of the paper, the results of our experiments on the Atari Hard exploration task, **including $Montezuma's Revenge$, can be found in Appendix E.** Since the main goal of our paper was to enable agents to learn in environments with noise sources (Noisy TV, stochasticity), not hard exploration tasks, we did not mentioned them in the paper. Furthermore, many studies that use intrinsic rewards and whose goal is not hard exploration tasks rarely conduct experiments on hard exploration tasks ([1], [4], etc.). Nevertheless, our proposed TeCLE has been experimented on hard exploration tasks and has shown better performance on average than [1], which is known to perform well on hard exploration tasks as well as in an environment with noise sources. We are sorry for the insufficient explanation.
>
> ---
> **Reference**
>
> [1] Burda, Yuri, et al. "Exploration by random network distillation." _arXiv preprint arXiv:1810.12894_ (2018).
>
> [2] Raileanu, Roberta, and Tim Rocktäschel. "Ride: Rewarding impact-driven exploration for procedurally-generated environments." _arXiv preprint arXiv:2002.12292_ (2020).
>
> [3] Pathak, Deepak, et al. "Curiosity-driven exploration by self-supervised prediction." _International conference on machine learning_. PMLR, 2017.
>
> [4] Mavor-Parker, Augustine, et al. "How to stay curious while avoiding noisy tvs using aleatoric uncertainty estimation." _International Conference on Machine Learning_. PMLR, 2022.

---

> > ### Comment · Reviewer_KD3W · 2024-11-24
> >
> > W1:
> > - Line 250-264: "In our intrinsic formulation, the generated εt+1 is used to sample the latent representation zt+1,
> > and the ϕˆ(st+1) is reconstructed from pψ using zt+1 and at. Therefore, it can be considered that
> > sequence ϕˆ(s(1:t)) = (ϕˆ(s1), · · · , ϕˆ(st)) has an amount of temporal correlation, depending on β.
> > We hypothesize that the temporal correlation and anti-correlation (β ̸= 0) in the generated noise
> > sequence determine the exploratory behavior of the agent. When temporally anti-correlated noise
> > with β < 0 is injected, noise sequences with constantly fluctuating magnitude can dynamically
> > produce the reconstructed state sequence. Thus, agents can be less sensitive to novel states, making
> > them more robust to Noisy TV by assigning smaller intrinsic rewards than when β ≥ 0. Besides, in
> > the injection of temporally correlated noise with β > 0, the noise sequence with smooth changing
> > magnitude generates a larger intrinsic reward in the novel states than when β ≤ 0. To be more
> > specific, temporally anti-correlated noise with β < 0 can make the proposed TeCLE continue to have
> > a perturbation of subsequent intrinsic rewards. On the other hand, the smooth change of temporally
> > correlated noise with β > 0 makes the change of subsequent intrinsic rewards stable. Therefore,
> > we expect that TeCLE can achieve higher performance with temporally correlated noise (β > 0)
> > in sparse reward environments and with temporally anti-correlated noise (β < 0) in environments
> > where Noisy TV exists."。--> These pretty much are how you implement your method. The only relevant piece to noisy TV is this sentence: "Thus, agents can be less sensitive to novel states, making them more robust to Noisy TV by assigning smaller intrinsic rewards than when β ≥ 0". I don't get the idea of this sentence. What do you mean by "less sensitive to novel states" and why assigning smaller intrinsic rewards make it robust to noisy TV problems.
> >
> > There are more examples of this kind of unclear writing. Here, I will give one more:
> > - Line 160: To solve the above issue, we propose a new intrinsic reward formulation called TeCLE, which consists of three parts, as shown in Figure 1. Similar to other curiosity-driven exploration methods, the intrinsic reward is computed separately from the policy networks. --> This starting sentence in Section 4 is followed by nothing but three sections of implementation details. I would suggest you communicate the main idea of why TeCLE can solve noisy TV problems in the beginning. At its current form, the readers have to dig into all three sections, searching those needles of "why TeCLE solves noisy problems" in the huge chunk of implementation details.
> >
> > - W2: I don't see why TeCLE is compatible with NovelID. Also, if the focus is noisy tv problems, this one discussed in the related works (https://proceedings.mlr.press/v162/mavor-parker22a/mavor-parker22a.pdf) should be considered as a baseline since they also provide a method to avoid noisy TV.
> >
> > I would keep my rating at this moment.

---

> ### Author Response · Authors · 2024-11-24
> **Reminder from Authors regarding the Discussion Period**
>
> Dear Reviewer,
>
> We would like to kindly remind you that we submitted our responses to your comments last Wednesday. As the discussion period is approaching its end, we sincerely ask for your attention to review our response. If you find that our answers sufficiently address your concerns and questions, we would be grateful if you could consider revising your rating.
>
> Thank you very much for your time and effort, and we truly appreciate your valuable feedback.

---

> ### Author Response · Authors · 2024-11-25
> **Rebuttal for Reviewer KD3W**
>
> We are grateful to the reviewers for their insightful feedback and constructive suggestions.
>
> ---
> *  **W1-1:** What do you mean by “less sensitive to novel states” and why assigning smaller intrinsic rewards make it robust to Noisy TV problems.
>
> *  **A1-1:** We appreciate the reviewers for their advice.
>
> In this rebuttal, we believed that we had sufficiently explained the impact of $\beta$ on NoisyTV. However, we apologize if you feel that the explanation was insufficient. Notably, we expect that the agent is less sensitive to new states is because $ϕ^​(s_{t+1}​)$ contains natural noise due to the perturbation, which is explained as:
>
> The problem with existing curiosity-based exploration methods is that they assign too large an intrinsic reward to new states. So when faced with a state like NoisyTV, where unpredictable new states are repeated, the agent will assign a large intrinsic reward and be trapped. However, the temporally anti-correlated ($\beta$<0) agents of TeCLE are robust to NoisyTV due to perturbations in their intrinsic rewards. In other words, as shown in Appendix A.2, color noise with $\beta$<0 has a large perturbation, **which could prevent the agent from consistently assigning large intrinsic rewards to the same unpredictable states.** Therefore, with such color noise injected into the latent space, TeCLE outputs $\hat{\phi}(s_{t+1})$ with perturbation to compute the intrinsic reward. The reason why **the agent** is less sensitive to new states is because $\hat{\phi}(s_{t+1})$ contains natural noise due to the perturbation. A TeCLE ($\beta$<0) trained in this way will not assign an excessively large intrinsic reward when faced with a state like NoisyTV.
>
> **&nbsp; &nbsp; &nbsp; &nbsp; A detailed explanation of why TeCLE is robust to NoisyTV is provided in Lines 253 - 267.**
>
> ---
>
> *  **W1-2:** I would suggest you communicate the main idea of why TeCLE can solve noisy TV problems in the beginning.
>
> *  **A1-2:** We appreciate the reviewers for their suggestions. In agreement with the reviewers, we will revise the beginning of the method in the paper. The existing version is in Lines 158-161 as follows:
>
>
>
> >“Although the existing curiosity-driven methods improved exploration, they can be vulnerable to Noisy TV problems or stochasticity of environments (Raileanu & Rockt¨aschel, 2020; Mavor-Parker et al., 2022). To solve the above issue, we propose a new intrinsic reward formulation called TeCLE, which consists of three parts, as shown in Figure 1. Similar to other curiosity-driven exploration methods, the intrinsic reward is computed separately from the policy networks.”
>
>
>
>
>
> &nbsp;  &nbsp;  &nbsp;  &nbsp; and the revised version will be as follows:
>
>
>
>
>
> >“Although the existing curiosity-driven methods improved exploration, they can be vulnerable to Noisy TV problems or stochasticity of environments (Raileanu & Rockt¨aschel, 2020; Mavor-Parker et al., 2022). TeCLE started with the assumption that this is caused by predicting the noise sources, which is inherently impossible, and the predictions themselves must contain noise to solve this problem. In the following paragraphs, we describe the role and effect of each part. Consequently, in part C. Colored noise, we prove that these effects ultimately help the agents deal with the Noisy TV problem. As shown in Figure 1, TeCLE consists of three parts and the intrinsic reward is computed separately from the policy networks.”
>
> ---
>
> *  **W2:** I don't see why TeCLE is compatible with NovelD. Also, AMA [1] should be considered as a baseline since they also provide a method to avoid Noisy TV.
>
> *  **A2:**  **Thanks for your sharp comments!!** NovelD is basically **a count-based method** similar to RIDE, so it cannot be compared to TeCLE. However, in large-scale stochastic environments, a prediction-based method such as RND [2] is used to calculate novelty in NovelD as follows: $novelty(s_t) = ||\phi(s_t)-\phi'_w(s_t)||_2$. Since **TeCLE is also a prediction-based method,** it can be calculated for novelty calculation in NovelD as follows: $novelty(s_t) = ||\phi(s_t)-\hat{\phi}(s_t)||_2$. Therefore, for the above two reasons, we think that TeCLE is applicable to NovelD, but the comparison is unsuitable. As in NovelD, the reason why we included AMA in related works but did not compare it is that TeCLE can be applied for AMA, and the comparison is unsuitable.
>
> ---
>
>
>
> **References**
>
> [1] Mavor-Parker, Augustine, et al. "How to stay curious while avoiding noisy tvs using aleatoric uncertainty estimation." _International Conference on Machine Learning_. PMLR, 2022.
>
> [2] Burda, Yuri, et al. "Exploration by random network distillation." _arXiv preprint arXiv:1810.12894_ (2018).

---

> > ### Comment · Reviewer_KD3W · 2024-11-26
> >
> > - W1: Now I'm getting the idea better now, but still have some questions.
> > My understanding now is that TeCLE injects noises to every state, so it will not assign overly high intrinsic rewards to noisy states. I think this will require the injected noises to have the same pattern with those in noisyTV states (e.g., both are Gaussian noises). Also, how do you know the intrinsic rewards on noisy states are large than the other clean states without noises? One possibility is that noisy states are associated with constantly small intrinsic rewards. Any evidence for this? Additionally, I think the explanation on Line 253-267 is very confusing and verbose. Your explanation in the rebuttal is better. I will suggest incorporating that to the paper.
> >
> > - W2:
> > I don't get why the distinction between prediction-based and count-based is the reason of not comparing with it. To me, RND and NovelD are methods of generating intrinsic rewards, and noisyTV robustness was discussed in their papers, so as a practitioner that uses intrinsic rewards, I would be curious how much TeCLE can improve over both methods in noisy TV problems.
> >
> > I still think you should compare it with AMA or integrate AMA with TeCLE. If AMA itself already did as well as TeCLE, then the significance of TeCLE will be undermined. On the other hand, to show that AMA is compatible with TeCLE, you need experiments to demonstrate it.
> >
> > As my question to W1 is partially addressed, I'll update my score.

---

> ### Author Response · Authors · 2024-11-28
>
> Thank you for the good point. In agreement with the reviewer's concern, we are performing experiments to compare TeCLE with AMA and NovelD.
>
> After the experiments are finished, **we will provide the rebuttal as soon as possible.**

---

> ### Author Response · Authors · 2024-12-01
> **Answer for the second comment of reviewer KD3W**
>
> Dear Reviewer KD3W, we are thankful for your detailed feedback and helpful recommendations. For a better understanding of our method by both you and the reviewers, we have conducted additional experiments, which caused a delay in providing our response. We apologize for this delay and hope you understand. Through these additional experimental results, we would like to demonstrate that the proposed method makes a significant contribution, as outlined below:
>
> ---
> *  **W1:** How do you know the intrinsic rewards on noisy states are larger than the other clean states without noises? One possibility is that noisy states are associated with constantly small intrinsic rewards. Any evidence for this?
>
>
>
> *  **A1:** We appreciate the reviewer's sharp points and suggestions. To prove that Noisy TV states are closely related to the amount of intrinsic reward, we have measured the intrinsic reward before and after the Noisy TV states. As we explained in Section 5.1, Noisy TV states, which display randomly generated pixels, occur when the agent chooses the 'done' action in Minigrid environments. As shown in the table below, **intrinsic reward from RND has increased nearly 50 times when a Noisy TV state occurs.** In contrast, **intrinsic reward from TeCLE has increased almost 1.8 times in the same situation.**
>
>
>
>
> | Intrinsic Reward | TeCLE | RND |
> |------------------|--------|------------|
> | Before Noisy TV | 0.0211 | 6.3695e-06 |
> | After Noisy TV | 0.0377 | 3.1671e-04 |
>
>
>
>
> &nbsp;  &nbsp;  &nbsp;  &nbsp; The above results lead to two meanings as follows:
>
> &nbsp;  &nbsp;  &nbsp;  &nbsp;  **Firstly, TeCLE avoids being trapped in Noisy TV states by assigning smaller intrinsic rewards.** This explains how TeCLE can outperform baselines despite it being a curiosity-based exploration method.
>
> &nbsp;  &nbsp;  &nbsp;  &nbsp;  **Secondly, TeCLE can continue to explore because it assigns larger intrinsic rewards than baselines.** Exploration is an important factor as the state space gets larger, and sufficient exploration must be performed for the agent to learn the environment. This demonstrates how TeCLE can outperform baselines while showing robustness about the Noisy TV states.
>
>
> ---
>
> *  **W2:** I still think you should compare it with AMA or integrate AMA with TeCLE.
>
>
> *  **A2:** In agreement with the reviewer's concerns, we have additionally compared TeCLE with AMA and NovelD in the Minigrid environments with additional experiments after receiving your second comments. To compare the performance, we measured the average returns of agents during training.
>
>
>
>
> | Environment | TeCLE | AMA | NovelD |
> |--------------|:-------:|:-----:|:--------:|
> | $DoorKey 8\times8$ | 0.96 |0.47 | **0.97** |
> | $DoorKey 16\times16$ | **0.98** |0.03 | 0.00|
> | $DoorKey 8\times8$ (Noisy TV) |0.96 | 0.42| **0.97**||
> | $DoorKey 16\times16$ (Noisy TV) |**0.98** | 0.03 | 0.00|
>
>
> &nbsp;  &nbsp;  &nbsp;  &nbsp; The table above shows that TeCLE achieved significantly higher performance than AMA in the experiments on $DoorKey8\times8$ and $DoorKey16\times16$. In particular, while AMA failed to learn the optimal policy network in the environments with NoisyTV, TeCLE succeeded in learning the optimal policy network. On the other hand, while NovelD showed higher performance than TeCLE by 0.01 in the $DoorKey8\times8$ environments, it failed to learn the policy networks in the $DoorKey16\times16$ at all. **It is notable that AMA and NovelD failed to learn in $DoorKey 16\times16$, where the state space is larger than $DoorKey 8\times8$. Therefore, we found that TeCLE has better exploration ability than AMA and NovelD.**
>
> ---
> &nbsp;  &nbsp;  &nbsp;  &nbsp; We hope that the additional experiments contribute to further validating the superiority of the proposed TeCLE. **If our additional experiments have adequately addressed the reviewers' concerns, we kindly ask you to consider this when evaluating the rating.** We would like to once again thank the reviewers for their feedback. Since the discussion period is still remaining, we hope to continue the discussion and address any remaining concerns.

---

### Official Review · Reviewer_2ARa · 2024-11-04

**Soundness:** 1
**Presentation:** 2
**Contribution:** 2
**Rating:** 3
**Confidence:** 3

**Summary:**

This paper aims to develop an approach to providing intrinsic rewards more robust to stochasticity (and noisy-TV scenarios) than previous approaches. The proposed approach–TeCLE–uses inverse dynamics modeling to learn a state representation and uses the action-conditioned state reconstruction loss (through a conditional VAE) as an intrinsic reward. While sampling the latent representation from the encoder’s output, temporally correlated noise is added (modulated by $\beta$). Empirical analysis shows that TeCLE outperforms bonuses such as RND and ICM in Minigrid environments with noisy-TV and some stochastic Atari environments. The authors also illustrate qualitative differences in exploration behavior based on the color of noise added.

**Strengths:**

S1. Using temporally correlated noise on a latent representation of inverse dynamics model features appears novel (but also see W2).

S2. The approach outperforms popular intrinsic bonuses like RND and ICM in Minigrid environments with noisy TVs and also in many of the considered atari environments.

**Weaknesses:**

W1. From the aggregated and normalized results in Figure 2, all noise levels seem to perform similarly (high overlap in standard errors), even white noise with $\beta=0$. This seems to indicate that colored/temporally correlated noise may not be so important to the performance of the proposed approach, which is a major theme of the paper.

W2. While the paper describes approaches that add noise in the action space, it misses comparisons with approaches that add noise in parameter space [1, 2], which would be closer to adding noise in some latent space. Similarly, another work adds temporally correlated noise to the latent state for exploration [3]. It would be helpful to compare the differences with the approach introduced here. Other forms of temporally correlated exploration, like temporally extended epsilon-greedy [4] could also be interesting to consider in the atari evaluation.

W3. The clarity of the paper requires improvement. Section 4A describes the inverse dynamics modeling task used for feature embedding. As far as I can tell, this is the same idea of feature embedding used in ICM. While the authors describe the ICM paper in the Background section, the current presentation of feature embedding in Section 4A is more likely to be incorrectly interpreted as a novel contribution of this paper. It should be clearly mentioned that the representation/feature learning part is the same as that of ICM. If there is a difference, please clarify what is new in that section.

Further, since ICM and TeCLE use the same feature embedding (which captures the same information about states), the difference in robustness to stochasticity between the two approaches must arise from the tasks used to generate intrinsic rewards (model error vs. reconstruction error), and it is not clear why reconstruction error would work better. Perhaps the authors could comment and elaborate on why TeCLE works better than ICM.

Please also see the “Questions” section for additional details that weren’t clear to me. I remain open to increasing the score should the weaknesses and questions be adequately addressed/clarified.

### References

[1] Meire Fortunato, Mohammad Gheshlaghi Azar, Bilal Piot, Jacob Menick, Ian Osband, Alex Graves, Vlad Mnih, Remi Munos, Demis Hassabis, Olivier Pietquin, et al. Noisy networks for exploration. arXiv preprint arXiv:1706.10295, 2017


[2] Rückstiess, T., Sehnke, F., Schaul, T., Wierstra, D., Sun, Y., & Schmidhuber, J. (2010). Exploring parameter space in reinforcement learning. Paladyn, 1, 14-24.

[3] Chiappa, A. S., Marin Vargas, A., Huang, A., & Mathis, A. (2023). Latent exploration for reinforcement learning. Advances in Neural Information Processing Systems, 36.

[4] Dabney, W., Ostrovski, G., & Barreto, A (2021). Temporally-Extended ε-Greedy Exploration. In International Conference on Learning Representations.

**Questions:**

- In Ablation Study 1, the authors show that the CVAE's action conditioning helps one of the considered problems but not the other. In lines 503-506, the authors mention that this is because of better self-supervised learning. It is not clear to me precisely what that statement means. Could the authors elaborate on why action conditioning is helpful?

- Is there a reason to use the Number of Updates as the x-axis in Figures 5/7 and the Number of Steps in Figures 6/8? Using the Number of Steps in all cases could be better, or specifying the correspondence between updates and steps (and frames in Atari) in the main text.

---

> ### Author Response · Authors · 2024-11-20
> **Rebuttal for Reviewer 2ARa**
>
> We are grateful to the reviewers for their insightful feedback and constructive suggestions.
>
> ---
> * **W1:** temporally correlated noise may not be so important to the performance of the proposed approach.
>
> * **A1:** Concerning W1, **we have added experiments with three additional Minigrid environments to Table 1:** $LavaCrossingS11N5$, $KeyCorridorS3R3$, and $MultiRoomN2S4$, and **modified Figure 2 in the paper.** The results of the another experiments in section 5 showed that temporally correlated/anti-correlated noise affects the exploration/exploitation performance of the agent. We also provided additional experiments on the $\beta$ of colored noise in Appendix D.
> &nbsp; &nbsp; &nbsp; &nbsp; In Figure 10, it can be seen that in environments with small state spaces, such as $Doorkey8\times8$ and $LavaCrossingS9N3$, all colored noise learns the optimal policy. However, as the state space becomes larger, such as $DoorKey16\times16$ and $LavaCrossingS11N5$, the difference in performance of the policy networks according to $\beta$ becomes obvious, and only some of the colored noise learns the optimal policy. **This indicates that the $\beta$ of the colored noise has a significant impact on the performance of the policy network.**
>
> ---
> * **W2:** The paper misses comparisons with other types of noise.
>
> * **A2:** Concerning W2, the studies mentioned can be compared to the proposed TeCLE in terms of applying temporal correlations to reinforcement learning.
> &nbsp; &nbsp; &nbsp; &nbsp; However, unlike studies that add noise to the parameter space [1, 2], add noise to the parameter and action space [3], and add randomness to the action selection [4], TeCLE does not inject noise into the action space. More specifically, studies that add noise to the parameter space encourage exploration by adding random noise to the parameters of the policy network. Furthermore, studies that add noise to the parameter and action spaces add noise to the parameters of the policy network and the sampled actions, and studies that add randomness to action selection add noise to the sampled actions. **Since these methods do not use intrinsic rewards, and TeCLE does not directly inject noise into the parameters, action space, or sampled actions, we compared them to exploration methods based on intrinsic rewards, which can be found in studies such as [5, 6, 7, 8].**
>
> ---
> * **W3:** The authors could comment and elaborate on why TeCLE works better than ICM.
>
> * **A3:** Concerning W3, TeCLE takes $\phi(s_{t+1})$ and $a_t$ as input and outputs $\hat{\phi}(s_{t+1})$. This is opposite to ICM, which takes $\phi(s_t)$ and $a_t$ as input and outputs $\hat{\phi}(s_{t+1})$. Furthermore, since TeCLE reconstructs the state based on the probability distribution, it can avoid being trapped by assigning too large an intrinsic reward to an unpredictable state. As a result, the proposed TeCLE achieved superior performance compared to ICM, which is a strong baseline, and showed robustness to the NoisyTV problem and stochasticity.
>
> ---
> * **Q1:** Could the authors elaborate on why action conditioning is helpful?
>
>
> * **A4:** For Q1, $DoorKey$ environment of Minigrid is one of the sparse reward environments, and the $DoorKey16\times16$ environment is a more difficult environment to learn because it has about twice the room size than the $DoorKey8\times8$ environment.
> &nbsp; &nbsp; &nbsp; &nbsp;We hypothesized that **the reason why only TeCLE with action was able to successfully learn the environment, as compared to TeCLE without action as a condition, in the $DoorKey16\times16$, this is because action allows for self-supervised learning,** which is similar to the arguments in [5, 9, 10, 11]. This was also mentioned in Lines 95-102.
>
> ---
> * **Q2:** Using the Number of Steps in all cases could be better, or specifying the correspondence between updates and steps (and frames in Atari) in the main text.
> * **A5:** For Q2, the experiment on Minigrid is 128 steps and one update per actor. As you pointed out, we have modified it by adding a description "The policy network is updated every 128 steps." on Line 377 and Appendix C.3 in the paper.

---

> ### Author Response · Authors · 2024-11-20
> **Reference - Rebuttal for Reviewer 2ARa**
>
> **Reference**
>
> [1] Meire Fortunato, Mohammad Gheshlaghi Azar, Bilal Piot, Jacob Menick, Ian Osband, Alex Graves, Vlad Mnih, Remi Munos, Demis Hassabis, Olivier Pietquin, et al. Noisy networks for exploration. arXiv preprint arXiv:1706.10295, 2017
>
> [2] Rückstiess, T., Sehnke, F., Schaul, T., Wierstra, D., Sun, Y., & Schmidhuber, J. (2010). Exploring parameter space in reinforcement learning. Paladyn, 1, 14-24.
>
> [3] Chiappa, A. S., Marin Vargas, A., Huang, A., & Mathis, A. (2023). Latent exploration for reinforcement learning. Advances in Neural Information Processing Systems, 36.
>
> [4] Dabney, W., Ostrovski, G., & Barreto, A (2021). Temporally-Extended ε-Greedy Exploration. In International Conference on Learning Representations.
>
> [5] Pathak, Deepak, et al. "Curiosity-driven exploration by self-supervised prediction." _International conference on machine learning_. PMLR, 2017.
>
> [6] Burda, Yuri, et al. "Exploration by random network distillation." _arXiv preprint arXiv:1810.12894_ (2018).
>
> [7] Badia, Adrià Puigdomènech, et al. "Never give up: Learning directed exploration strategies." _arXiv preprint arXiv:2002.06038_ (2020).
>
> [8] Guo, Zhaohan, et al. "Byol-explore: Exploration by bootstrapped prediction." _Advances in neural information processing systems_ 35 (2022): 31855-31870.
>
> [9] Yuri Burda, Harri Edwards, Deepak Pathak, Amos Storkey, Trevor Darrell, and Alexei A Efros. Large-scale study of curiosity-driven learning. arXiv preprint arXiv:1808.04355, 2018a.
>
> [10] Deepak Pathak, Dhiraj Gandhi, and Abhinav Gupta. Self-supervised exploration via disagreement. In International conference on machine learning, pp. 5062–5071. PMLR, 2019.
>
> [11] Roberta Raileanu and Tim Rockt¨aschel. Ride: Rewarding impact-driven exploration for procedurally-generated environments. arXiv preprint arXiv:2002.12292, 2020.

---

> ### Author Response · Authors · 2024-11-24
> **Reminder from Authors regarding the Discussion Period**
>
> Dear Reviewer,
>
> We would like to kindly remind you that we submitted our responses to your comments last Wednesday. As the discussion period is approaching its end, we sincerely ask for your attention to review our response. If you find that our answers sufficiently address your concerns and questions, we would be grateful if you could consider revising your rating.
>
> Thank you very much for your time and effort, and we truly appreciate your valuable feedback.

---

> > ### Comment · Reviewer_2ARa · 2024-11-25
> >
> > Thank you for your response. I appreciate the efforts to improve the paper in the rebuttal phase.
> >
> > Overall, my concerns with this paper remain largely unresolved. Even though some of the results are indeed interesting, the motivation behind many of the ideas is still unclear – temporally correlated noise, action-conditioning, and why reconstruction is better than prediction in a latent space (that tries to remove uncontrollable factors). Noting that these ideas sometimes improve empirical performance is insufficient without understanding/communicating the conceptual reasons behind the results. Further, I am not sure why the authors don't consider other approaches that use temporally correlated noise in the analysis; just because they do not use an intrinsic bonus does not seem like a satisfying reason.

---

> ### Author Response · Authors · 2024-11-26
> **Rebuttal for Reviewer 2ARa**
>
> We are grateful to the reviewers for their insightful feedback and constructive suggestions. We are sorry that you were unable to clearly identify the differences between the existing method and our proposed approach of maintaining a large intrinsic reward. Therefore, we have prepared the following for a better understanding as:
>
>
>
> In agreement with the reviewer's concern, we have added explanations and additional experimental results to Appendix F.
>
> We have compared the intrinsic rewards of TeCLE and those of baselines to explain how TeCLE can be robust to noise sources while outperforming baselines.
>
> **Notably, as shown in Figure 16, while the measured intrinsic reward of the baselines shows a small value near zero, the measured intrinsic reward of TeCLE maintains a relatively large value.**
>
> The experimental results prove our hypothesis that TeCLE can be robust to noise sources since it continuously injects noise when reconstructing state representation. **Therefore, unlike baselines that maintain smaller intrinsic rewards since they minimize the prediction error of the state representation, TeCLE maintains a large intrinsic reward since it contains noise regardless of whether it is sufficiently explored.**
>
> As a result, this tendency of intrinsic reward from TeCLE helps agents prevent being trapped in environments that contain inherently unpredictable noise sources.
>
> **We hope that the reviewers can understand what we have proposed through the differences shown in the graph (Figure 16 in Appendix F) representing the intrinsic reward generated by the existing method and the proposed TeCLE.** We also expect that we can address more questions during the extended discussion period.
>
> Thank you for your valuable feedback.

---

> ### Author Response · Authors · 2024-11-26
> **Gentle Reminder for the Rebuttal for Reviewer 2ARa**
>
> Dear Reviewer 2ARa,
>
> We have attached the answer for your feedback. However, there was no email forwarding for it. So we want to remind reviewers for our rebuttal, where we have added new Appendix F including a graph that shows the difference between intrinsic rewards from the existing works and proposed TeCLE. Please check the below rebuttal for Reviewer 2ARa and revised PDF including new Appendix F. Thanks for your feedback.

---

### Author Response · Authors · 2024-11-22
**Request for Review of the Rebuttal**

Dear Reviewers,

We have prepared this rebuttal by addressing your comments from two days ago. We understand you have many commitments and a busy schedule, but we hope to discuss our responses to your comments. If our rebuttal sufficiently addresses your concerns, we kindly ask you to reconsider your ratings.

Thank you.

---

### Comment · Area_Chair_Qgu2 · 2024-11-24

Dear Reviewers,

thank you for your service. As the deadline for the discussion period is approaching, I strongly encourage you to submit your response to the authors soon. At least, an acknowledgment of having read the rebuttal should be submitted.

Best regards,

AC

---

### Author Response · Authors · 2024-11-27
**Gentle Reminder for deadline of PDF modification**

Dear Reviewers,

As the deadline for finalizing the revised version of the paper's PDF approaches, further modifications to the PDF are no longer possible. If reviewers raise additional discussions or concerns, we will do our best to address them thoroughly in the rebuttal. To respond to the concerns raised by the four reviewers, we have added several appendices and made revisions and additions to the figures and tables.

Please review these sections, and **if you find that the concerns you raised have been adequately addressed, we kindly ask you to reconsider your rating.**

Once again, we sincerely thank the reviewers for their valuable feedback.

---

### Author Response · Authors · 2024-12-03
**Final comment for all reviewers**

We would like to thank the reviewers for their comments and help in improving the paper. As the discussion period for the paper is nearing its end, we have written the following message to encourage reconsideration and reflection on any misunderstandings regarding our contributions and achievements.



In this paper, we proposed a novel curiosity-based intrinsic reward method, TeCLE.



In our proposed method, the intrinsic reward is calculated based on **the state reconstructed by the cVAE,** and the exploratory behavior of the agent is determined by applying colored noise to the cVAE. Extensive experiments on Atari and Minigrid environments show that TeCLE has good exploration abilities and is robust to NoisyTV.



---



**Common concerns of reviewers with the TeCLE proposed in this paper are as follows:**





* It is not clear why reconstruction error and temporal correlation are better than prediction error (ICM, RND).





* Limited comparison between other methods.





* Clarification is needed on how color noise determines the exploratory nature of the agent.





* Contribution is limited.



However, we have tried to show that the proposed TeCLE has the following contributions and outstanding works as:



---

Firstly, in agreement with the reviewer’s concerns, we have revised the paper. **A comparison between the draft and revised versions of the submitted paper is as follows:**



* We have added experiments with **three additional Minigrid environments** to Table 1, thereby revising Figure 1.

* We have supplemented the Related Works section in the revised version of the paper.

* We have conducted **additional experiments** and **Appendix F** has been added, providing a comparison of intrinsic rewards with baselines to analyze how TeCLE outperforms baselines while successfully learning optimal policy networks.

* We have added the **description for the Minigrid experiment results** in the revised version of the paper.

* We have modified some ambiguous sentences and statements, which were pointed out by reviewers, to be clearer.



---




Besides, while not in the revised version of the paper, here are some experiments we conducted during the review period to address reviewer concerns and the results of those efforts:



* We have compared **the intrinsic rewards of RND and TeCLE before and after the Noisy TV states.** This explains why TeCLE is robust against Noisy TV and has achieved relatively good scores even in the most sparse compensation environments, as opposed to baselines (for a detailed explanation, please refer to the rebuttal for reviewer KD3W).

* We have measured the computational overhead of ICM, RND, and TeCLE. **The results clearly show that TeCLE has less computational overhead compared to ICM and RND.** Regarding the model structure, we chose CNN over MLP in the cVAE since it enables more efficient processing and better abstraction. To further reduce computational cost, we minimized the number of channels in convolutional layers. These designs were made to achieve a balance between efficiency and performance (for a detailed explanation, please refer to the rebuttal for reviewer GN7S).

* We have conducted **additional experiments to compare TeCLE with AMA and NovelD,** which are the strong baselines. Experimental results show that TeCLE outperforms AMA and NovelD (for a detailed explanation, please refer to the rebuttal for reviewer KD3W).


---
**We hope that the answers and further experiments during the review period helped address your suggestions and concerns. We sincerely thank you all reviewers.** Thanks again.



Best Regards,

Authors

---

### Meta-Review · Area_Chair_Qgu2 · 2024-12-17

**Metareview:**

This paper introduces TeCLE, a new method for improving exploration in reinforcement learning by incorporating temporally correlated noise into the intrinsic reward. The core idea involves using a conditional VAE to reconstruct state embeddings, with colored noise injected during latent sampling. This approach aims to address the "noisy TV problem" and enhance robustness to stochasticity in environments.

Strengths
-----------
- **Novelty:** The use of temporally correlated noise within a VAE-based intrinsic reward mechanism appears to be a novel contribution.
- **Performance:** Empirical results suggest TeCLE outperforms established baselines like ICM and RND in certain Minigrid and Atari environments, particularly those with noisy TVs.

Weaknesses
--------------
- **Limited contribution and motivation:** Several reviewers argue that the core contribution of injecting colored noise into the VAE latent space is somewhat incremental. The motivation behind using colored noise and its connection to improved exploration is not clearly articulated.

- **Empirical evaluation:** Concerns are raised about the sufficiency of the empirical evaluation. Specifically: missing confidence intervals and baseline results in Table 1, lack of clarity regarding hyperparameter selection for baselines, absence of key baselines and challenging exploration tasks like Montezuma's Revenge.

Moreover, Reviewers expressed concerns about the quality of writing, inaccurate statements, and questionable claims.

Reviewers agree that TeCLE presents an interesting idea with some promising empirical results; however, the paper suffers from several weaknesses. The novelty and impact of the contribution appear limited, and the motivation and theoretical grounding require further elaboration.  The empirical evaluation needs strengthening through the inclusion of missing information, additional baselines, and more challenging environments. I encourage the Authors addressing the Reviewers' concerns in a future submission.

**Additional Comments On Reviewer Discussion:**

The Authors' rebuttal attempted to address most of Reviewers' concerns about the motivation of the paper and the quality of the experimental evaluation. Three Reviewers out of four have replied, with the majority keeping their score.

---

### Decision · Program_Chairs · 2025-01-22

Reject